# Partition and Code: learning how to compress graphs

**Giorgos Bouritsas**[*]
Imperial College London, UK
g.bouritsas@imperial.ac.uk

**Andreas Loukas**
EPFL, Switzerland
andreas.loukas@epfl.ch

**Nikolaos Karalias**
EPFL, Switzerland
nikolaos.karalias@epfl.ch

**Michael M. Bronstein**
Imperial College London / Twitter, UK
m.bronstein@imperial.ac.uk

## Abstract

Can we use machine learning to compress graph data? The absence of ordering in graphs poses a significant challenge to conventional compression algorithms, limiting their attainable gains as well as their ability to discover relevant patterns. On the other hand, most graph compression approaches rely on domain-dependent handcrafted representations and cannot adapt to different underlying graph distributions. This work aims to establish the necessary principles a lossless graph compression method should follow to approach the entropy storage lower bound. Instead of making rigid assumptions about the graph distribution, we formulate the compressor as a probabilistic model that can be learned from data and generalise to unseen instances. Our "*Partition and Code*" framework entails three steps: first, a partitioning algorithm decomposes the graph into subgraphs, then these are mapped to the elements of a small dictionary on which we learn a probability distribution, and finally, an entropy encoder translates the representation into bits. All the components (partitioning, dictionary and distribution) are parametric and can be trained with gradient descent. We theoretically compare the compression quality of several graph encodings and prove, under mild conditions, that PnC achieves compression gains that grow either linearly or quadratically with the number of vertices. Empirically, PnC yields significant compression improvements on diverse real-world networks.[1]

## 1 Introduction

Lossless data compression has been one of the most fundamental and long-standing problems in computer science. It is by now well-understood that the intrinsic limits of compression are governed by the entropy of the underlying data distribution [1]. Crucially, these limits expose an intimate connection between compressibility and machine learning: the better one models the underlying data distribution (from limited observations) the more bits can be saved and vice-versa [2].

The compression of ordered data such as text, images, or video, underpins the modern technology from web protocols to video streaming. However, graph-structured data remain a notable exception. As graph data are becoming more prevalent, it becomes increasingly important to invent practical ways to encode them parsimoniously.

There are three main challenges one faces when attempting to compress graphs:

---

[*]This work was done while Giorgos Bouritsas was visiting Dr. Andreas Loukas at EPFL, Lausanne.
[1]The source code is publicly available at https://github.com/gbouritsas/PnC

35th Conference on Neural Information Processing Systems (NeurIPS 2021).

*C1. Dealing with graph isomorphism (GI).* A key difficulty that distinguishes graphs from conventional data lies in the absence of an inherent ordering of the graph vertices. In order to be able to approach the storage lower bounds, isomorphic graphs should be encoded with the same codeword. However, since the complexity of known algorithms for GI is super-polynomial on the number of vertices [3], a direct use of GI is impractical for graphs consisting of more than a few hundred vertices. Indeed, an examination of the graph compression literature reveals that most progress has been made by optimising a vertex ordering and adapting methods originally invented for vectored data [4–6]. Unfortunately, naively encoding graphs as vectors results in a significant loss in compression[2].

*C2. Evaluating the likelihood.* An optimal encoder [1] requires one to accurately estimate and evaluate the probabilities of all the possible outcomes of the underlying domain. When dealing with high-dimensional data, this can be addressed by partitioning the data into parts to obtain a decomposition of the probability distribution: e.g., images can be compressed by modelling the distribution of pixels or patches [10–12], and text by focusing on characters or n-grams [13–19]. However, since graphs do not admit an efficiently computable canonical ordering, it is unclear what decomposition one should employ.

*C3. Accounting for the description length of the learned model.* The classical learning theory trade-off between model complexity and generalisation is of paramount concern for effective compression. Though in typical deep learning applications one can aim to model the data distribution with an overparametrised neural network (NN) that generalises well, utilising such models to compress information is problematic: since decoding is impossible unless the decoder also receives the learned model (i.e., the NN parameters), *overparametrised models are, by definition, suboptimal*. This is a pertinent issue for likelihood-based neural approaches as overparametrisation is commonly argued to be a key component of why NNs can be trained [20–22].

**The Partition and Code (PnC) framework.** Our main contribution is PnC, a framework for learning compression algorithms suitable for encoding graphs sampled from an underlying distribution. In the heart of PnC lie two ideas: *(a) Learn to break the problem into parts.* Rather than predicting directly the likelihood, we aim to learn how to decompose graphs into non-overlapping subgraphs, see Fig. 1. *(b) Identify and code recurring subgraphs when possible.* We use a learned *dictionary* to code subgraphs that appear frequently, whereas rare subgraphs are encoded separately. The dictionary is restricted to contain only a finite number of small recurring subgraphs. This biases the model towards interpretable and well generalising solutions. Both the "partition" and "code" components of PnC are learned directly from the data, by optimising the total description length.

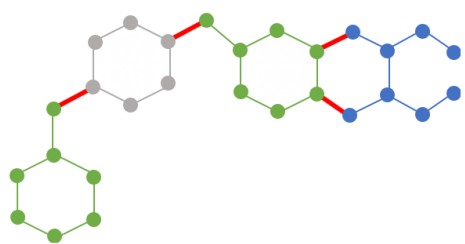

Figure 1: Illustration of the graph decomposition. The subgraph colours correspond to dictionary atoms $a_1$, $a_2$ and $a_3$. Cuts are denoted in red.

Our framework provides a solution to the three challenges of graph compression. C1: By constraining the dictionary to only contain graphs of size up to a small constant, we can efficiently solve GI. C2: Graph partitioning provides us with the desirable decomposition of the distribution. Using appropriate parametrised probabilistic models, we obtain a closed-form expression for the likelihood that can be later used by any close-to-optimal entropy coder [23–25]. C3: Since we use NNs to *decompose* the distribution (and not to predict the likelihood), we can rely on overparametrisation without having to relay the NN parameters to the decoder. Also, the complexity of our learned hypothesis (the dictionary) can be computed, and thus optimised, during training.

**Theoretical results.** Our analysis reveals that PnC can significantly improve upon less sophisticated graph encoders and justifies the usefulness of both the "Partition" and the "Code" component. Specifically, we prove that under mild conditions on the underlying graph distribution, PnC requires in expectation $\Theta(n^2)$ less bits than standard graph encodings, even if the latter are given access to an

---

[2]This follows by a simple counting argument: there are $2^{\binom{n}{2}}$ labelled undirected graphs while the respective number for unlabelled graphs is asymptotically equal to $2^{\binom{n}{2}}/n!$ [7]. Thus, if all graphs of $n$ vertices are equally probable, an encoding that does not consider isomorphism would sacrifice $\log n!$ bits [8, 9].

oracle that solves GI. Further, the dictionary induces additional savings of $\Theta(n)$ bits, with the gain being inversely proportional to the entropy of the distribution of the dictionary atoms. Thus, the more repetitive the patterns in the graph distribution are, the larger will be the compression benefits of PnC.

**Practical algorithms.** We instantiate of our framework using the following algorithmic modules: (a) a low-parameter learnable estimate of the probability distribution, (b) learning to select the dictionary from a graph universe of finite size, and (c) learning to partition. The latter is a parametric randomised iterative algorithm, the probabilities of which are inferred from a Graph Neural Network (GNN) and optimised with reinforcement learning. Importantly, all algorithms can be jointly trained in order to minimise the total description length in a synergistic manner. We evaluate our framework on diverse real-world graph distributions and showcase compression gains with respect to both conventional and advanced baseline compressors, in observed and unseen data.

## 2 Related work

**Engineered codecs.** The majority of graph compressors are not probabilistic, but rely on hand-engineered encodings optimised to take advantage of domain-specific properties of e.g., Web-Graphs [4], social networks [6, 26, 27], and biological networks [28]. A common idea in these approaches is *vertex reordering* [4–6, 26, 27, 29–31], where the adjacency matrix is permuted in such a way that makes it "compression-friendly" for mainstream compressors of sequences, such as gzip. The algorithms identifying the re-orderings are usually based on heuristics taking advantage of specific network properties, e.g., community structure. Another recurrent idea is to detect or use predefined *frequent substructures* (e.g., cliques) to represent more efficiently different parts of the graph via grammar rules [32]. These approaches do not attempt to model the underlying graph distribution and thus to approach the storage lower bounds, but strive to find a balance between compression ratios and fast operations on the compressed graphs. Thus, despite their practical importance, they are less relevant to our work. A comprehensive survey can be found in [33].

**Theory-driven approaches.** Several works have contributed to the foundations of the information content and the complexity of graphs [8, 9, 34–40]. However, few works have attempted to model the underlying graph distribution. Perhaps the most outstanding progress has been made for graphs modelled by the Stochastic Block Model (SBM) [41–48]. Although originally invented for clustering and network analysis purposes, these approaches can be seamlessly used for compression due to their exact likelihood computation. In fact, as we argue in this work, virtually any graph clustering algorithm can be used successfully for compression, by defining codewords corresponding to a community-based random graph model. However, as our experiments confirm, such approaches are less effective at compressing graphs that do not contain clusters.

**Likelihood-based neural approaches.** Any generative model that can provide likelihood estimates in a finite sample space can be used for lossless compression. As a result, a plethora of likelihood-based neural compressors have been recently invented, ranging from autoregressive models for text [18, 19, 49, 50] and images [11, 12] to latent variable models [51–55] (paired with bits-back coding [56, 57] and Asymmetric Numeral Systems - ANS [25]), normalising flows [58–61] and most recently, diffusion-based generative models [62, 63]. However, the vast majority of current graph generators lack the necessary theoretical properties an effective graph compressor should have: they compute probabilities on labelled graphs instead of isomorphism classes by resorting to a heuristic ordering [64–71] (in general this will be suboptimal unless we canonicalise the graph/solve graph isomorphism, while different orderings will have non-zero probabilities, hence we will incur compression losses), and/or do not provide a likelihood [72–76].

An important caveat, which is often ignored in the literature, is that when parametrising the distribution with a neural network, the data cannot be recovered unless the decoder has access to the network itself. Hence it is necessary to account for the NN's description length when evaluating compression gains. However, generative models are usually parameter inefficient, while compressing them (especially during training) is a challenging problem [77]. Note that this is significantly different than compressing neural classifiers, since the capacity to infer the likelihood up to high precision needs to be retained. Therefore, these approaches can have diminishing (or even negative) returns when the dataset size is not large enough. In contrast, by optimising the total description length (equivalent to *maximum a-posteriori*), we design a compressor that is practical even for small datasets, while the learned dictionary makes our compressor interpretable.

Other related work includes using compression objectives paired with heuristic algorithms for downstream tasks, such as motif finding [78, 79] and graph summarisation [80, 81], lossy compression/coarsening [82–87], and graph dictionary learning in the context of sparse coding [88, 89].

## 3 Preliminaries

We use capital letters for sets $A, B$, bold font for vectors $\mathbf{x}$ and matrices $\mathbf{X}$, and calligraphic font $\mathcal{X}$ for families of sets. We express the information content in bits by using a base-2 logarithm $\log$.

**Graphs.** Let $G = \{V, E\}$ be a graph with $n$ vertices $V = \{v_1, \ldots, v_n\}$ and $m$ edges, where $e_{ij} \in E$ whenever vertices $v_i, v_j \in V$ are connected by an edge. For simplicity, we assume the graphs to be undirected, though the same methods apply to directed graphs as well with minimal modifications. Let $\mathrm{cut}(A, B) = \{e_{ij} \in E \mid v_i \in A \text{ and } v_j \in B\}$ denote the set of edges with endpoints at two different vertex sets $A, B \subseteq V$.

**Information theory.** Following the usual terminology in information theory and Minimum Description Length (MDL) theory [2, 90, 91], we assume an *observation space* $\mathfrak{G}$ (in our case a space of possible graphs), and a probability distribution $p$ (sometimes referred to as *probabilistic source*) producing samples from the observation space. We observe a dataset $\mathcal{G} = \{G_1, G_2, \ldots, G_{|\mathcal{G}|}\}$ of i.i.d. observations drawn from $p$. Note that this setting is in contrast with most works on graph compression [4, 30, 5, 6], where the target is to compress a *single* large network, such as a social network or a web graph. Let a *description method* or *symbol code* $\mathsf{CODE} : \mathfrak{G} \to \{0, 1\}^*$ be a mapping from the observation space to a variable-length sequence of binary symbols, the output of which is a *codeword*.

In the context of graph compression, we are interested in the description length of the code $\mathrm{L}_{\mathsf{CODE}}(G)$, or $\mathrm{L}(G)$ for brevity, i.e., the number of bits needed to encode the graph $G$, rather than the code itself, given a single requirement: the code needs to be *uniquely decodable*, meaning that any concatenation of codes can be uniquely mapped to a sequence of observations. This property can be easily verified using only the description lengths by the well-known *Kraft–McMillan inequality* (l.h.s. in formula (1)), an important implication of which is that every uniquely decodable code implies a probability distribution $q(G)$ (r.h.s):

$$\sum_{G \in \mathfrak{G}} 2^{-\mathrm{L}(G)} \leq 1 \text{ (code)} \iff q(G) = \frac{2^{-\mathrm{L}(G)}}{\sum_{G \in \mathfrak{G}} 2^{-\mathrm{L}(G)}} \text{ (distribution)} \tag{1}$$

This equivalence allows us to always define underlying probabilistic models when working directly with description lengths. When the Kraft–McMillan inequality holds with equality, we say that the code is *complete*; the case of strict inequality means the code is *redundant*. Moreover, in the case of complete codes, the expected length of the code corresponds to the distribution entropy $\mathbb{H}_q[G]$ (plus a constant term when the code is redundant). Thus, compression is synonymous to defining probabilistic models with the lowest possible entropy. An important notion that frequently appears in our theoretical analysis is the *binary* entropy, i.e., the entropy of a bernoulli variable with probability of success $p$. To distinguish it from the general notion of entropy we will denote it as $\mathrm{H}(p) = -p \log p - (1 - p) \log(1 - p)$. It holds that $0 \leq \mathrm{H}(p) \leq 1$, where the l.h.s equality is satisfied for $p \in \{0, 1\}$ and the r.h.s. for $p = 1/2$.

**Common graph encodings.** An important principle that we follow is that whenever we cannot make any assumptions about an underlying distribution, or modelling it is impractical, then the uniform distribution $p_{\mathrm{unif}}$ is chosen for encoding. The reason is that uniform distribution is *worst-case optimal* [90]: for any non-uniform unknown distribution $p$, there always exists a distribution $q$ the corresponding encoding of which will be on expectation worse than the uniform encoding: $\mathbb{E}_{x \sim p}[-\log q(x)] > \mathbb{E}_{x \sim p}[-\log p_{\mathrm{unif}}(x)]$. In the context of graphs, when we cannot make any assumptions or when enumeration is impossible, we will be using a slightly more informative distribution: the Erdős–Rényi (ER) random graph model. This model assigns equal probability to all labelled graphs with $n$ vertices and $m$ edges. Assuming a uniform probability over the possible number of vertices $n$ and number of edges $m$ given $n$, we get:

$$L_{\mathrm{null}}(G) = \log(n_{\max} + 1) + \log\left(\binom{n}{2} + 1\right) + \log\binom{\binom{n}{2}}{m}, \tag{2}$$

where $n_{\max}$ is an upper bound on the number of vertices. The "*null*" encoding compresses more efficiently graphs that are either very sparse or very dense (low-surprise) as the number of possible graphs with $m$ edges is maximised when $m = n(n - 1)/4$.

# 4 The Partition & Code (PnC) graph compression framework

Our pipeline consists of three main modules: a *partitioning* module, a *dictionary* module, and an *entropy encoding* module. (a) The partitioning module is responsible for decomposing the graph into disjoint subgraphs and cross-subgraph edges (or *cuts*). Subgraphs play the role of elementary structures, akin to characters or words in text compression, and pixels or patches in image compression. (b) The dictionary is a small collection of subgraphs (*atoms*) that are recurrent in the graph distribution. The dictionary module maps the partitioned subgraphs to atoms in the dictionary, allowing us to represent the graph as a collection of atom indices and cuts. (c) Finally, this representation is given as input to the an entropy encoder that translates it into bits.

Decoding the graph in a lossless way involves inverting these three steps: initially the atom indices and cuts are decoded using the same probabilistic model with the encoder, then the atoms are retrieved from the dictionary, and finally all elements are composed back to obtain the original structure. The composition becomes possible by making sure that the cuts are encoded w.r.t. an arbitrarily chosen ordering of each atom's vertices, hence the decoded graph is guaranteed to be isomorphic to the input, but not necessarily with the same vertex ordering.

## 4.1 Step 1. Partitioning

The first step of PnC is to employ a parametric partitioning algorithm $\mathsf{PART}_\theta$ to decompose each graph $G$ into $b$ subgraphs of bounded size:

$$\mathsf{PART}_\theta(G) = (\mathcal{H}, C) \quad \text{and} \quad \mathcal{H} = \{H_1, H_2, \cdots, H_b\}, \tag{3}$$

where $H_i = \{V_i, E_i\}$ is the $i$-th subgraph and $C = \{V, E_C\}$ is a $b$-partite graph containing all cut edges $E_C = E - \cup_i E_i$. Variable $\theta$ indicates the learnable parameters.

*Is partitioning necessary?* In order to convert probability estimates to codewords, entropy encoders [23–25] need to be able to compute the probability of *every graph in the observation space* (or equivalently to have access to the cumulative distribution function (c.d.f.)) - see challenge C2 in Section 1. To achieve this goal, one needs to partition the observation space in a way that permits efficient enumeration of the possible outcomes. Analogously, as we will see in Section 4.3, graph partitioning allows us to decompose the distribution and thus to obtain a closed-form expression for the likelihood. Further, Theorem 1 suggests that partitioning brings a useful inductive bias for graph data providing significant storage gains compared to distribution-agnostic baselines.

## 4.2 Step 2. Graph dictionary

Rather than naively compressing each subgraph in (3) under a null model, an effective compression algorithm should exploit regularities in the output of $\mathsf{PART}_\theta$. We propose to utilise a dictionary that stores the most commonly occurring subgraphs. Concretely, we define a dictionary $D$ to be a collection of connected subgraphs (or *atoms*) from some universe $\mathfrak{U}$:

$$D = \{a_1, a_2, \cdots, a_{|D|}\}, \quad \text{where} \quad a_i \in \mathfrak{U}. \tag{4}$$

There are two viable choices for the atom encoding: (a) If the universe is small enough to be efficiently enumerable then we can assume a uniform distribution over $\mathfrak{U}$ which yields the description length $\mathrm{L}(D) = |D| \log |\mathfrak{U}|$. Intuitively, this would amount to storing the index of each atom within a list enumerating $\mathfrak{U}$. (b) On the other hand, when $\mathfrak{U}$ is too large to enumerate the atoms can be stored one-by-one based on the null-model encoding given in (2):

$$\mathrm{L}(D) = \sum_{a_i \in D} \mathrm{L}_{\text{null}}(a_i). \tag{5}$$

It is important to note that, to be as effective as possible, the partitioning and the dictionary should be co-designed: $D$ should capture those subgraphs that are most likely, whereas $\mathsf{PART}_\theta$ should be biased towards subgraphs with similar structure. The use of a dictionary makes it possible to explicitly account for (and thus optimise) the description length of the learned hypothesis (i.e., equation above), which is an essential component of any compression algorithm (see challenge C3 in Section 1).

### 4.3 Step 3. Graph encoding

The last step entails compressing $G$ by encoding the output $(\mathcal{H}, C)$ of $\mathsf{PART}_\theta$. As discussed in Section 3, a uniquely decodable code implies a probability distribution, i.e., $q_\phi(G) = q_\phi(\mathcal{H}, C)$ which corresponds to a description length $\mathrm{L}_\phi(\mathcal{H}, C) = -\log q_\phi(\mathcal{H}, C)$, where $\phi$ denotes the learnable parameters. In the following we explain how $q_\phi$ is decomposed.

**Subgraphs.** We opt for a dual encoding of subgraphs: one for the subgraphs that belong to the dictionary $\mathcal{H}_{\mathrm{dict}}$, and one for non-dictionary subgraphs $\mathcal{H}_{\mathrm{null}}$ that are encoded with the help of a null model as in Eq. (2). This choice has a dual purpose: (a) It allows $\mathsf{PART}_\theta$ to choose non-dictionary atoms. This is crucial to our approach, since constraining the partitioning to specific isomorphism classes would significantly complicate optimisation. (b) Further, it enables us to maintain a balance between two common structures found in real-world networks—*frequent subgraphs* (stored in the dictionary), and *low-entropy subgraphs* as implied by the null model (i.e., very sparse or very dense subgraphs). The distribution is therefore decomposed into the following components:

*Number of subgraphs.* First, we encode the number of dictionary and non-dictionary subgraphs ($b_{\mathrm{dict}}$ and $b_{\mathrm{null}}$ respectively) as follows:

$$q_\phi(b_{\mathrm{dict}}, b_{\mathrm{null}}) = \mathrm{Binomial}(b_{\mathrm{dict}}|b; \phi)q_\phi(b) = \binom{b}{b_{\mathrm{dict}}}(1 - \delta_\phi)^{b_{\mathrm{dict}}}\delta_\phi^{b - b_{\mathrm{dict}}}q_\phi(b), \qquad (6)$$

where $1 - \delta_\phi = \mathbb{P}[H \in D]$ is the probability of an arbitrary subgraph to belong in the dictionary and $q_\phi(b)$ is a categorical.

*Dictionary subgraphs.* The dictionary subgraphs are encoded in a permutation invariant way via a *multinomial* distribution, i.e., we encode the histogram of atoms:

$$q_\phi(\mathcal{H}_{\mathrm{dict}}|b_{\mathrm{dict}}, D) = \mathrm{Multinomial}(b_1, b_2, \ldots, b_{|D|} \mid b_{\mathrm{dict}}; \phi) = b_{\mathrm{dict}}! \prod_{a \in D} \frac{q_\phi(a)^{b_a}}{b_a!}, \qquad (7)$$

where $b_a = \left|\{H \in \mathcal{H}_{\mathrm{dict}}|H \cong a\}\right|$ and $\sum_{a \in D} b_a = b_{\mathrm{dict}}$.

*Non-Dictionary subgraphs.* The non-dictionary subgraphs are encoded independently according to the null model:

$$q_\phi(\mathcal{H}_{\mathrm{null}}|b_{\mathrm{null}}, D) = \prod_{H_i \in \mathcal{H}_{\mathrm{null}}} q_{\mathrm{null}}(H_i). \qquad (8)$$

**Cuts.** We encode the cuts conditioned on the subgraphs, using a non-parametric uninformative null model for multi-partite graphs similar to [45] (see Appendix B.1 for the detailed expression) that prioritises low-entropy cuts. In this way we give more emphasis to the subgraphs and an inductive bias towards distinct clusters in the graph.

Overall, the description length of $(\mathcal{H}, C) = \mathsf{PART}_\theta(G)$ is given by

$$\mathrm{L}_\phi(\mathcal{H}, C|D) = \mathrm{L}_\phi(b_{\mathrm{dict}}, b_{\mathrm{null}}) + \mathrm{L}_\phi(\mathcal{H}_{\mathrm{dict}}|b_{\mathrm{dict}}, D) + \mathrm{L}_{\mathrm{null}}(\mathcal{H}_{\mathrm{null}}|b_{\mathrm{null}}, D) + \mathrm{L}_{\mathrm{null}}(C|\mathcal{H}), \qquad (9)$$

and the learnable parameter set is $\left\{\delta_\phi, \{q_\phi(b)\}_{b=b_{\min}}^{b_{\max}}, \{q_\phi(a)\}_{a \in D}\right\}$.

**Remarks about graph isomorphism (GI).** Observe that given a fixed decomposition, our parametrisation is invariant to isomorphism, which is a desirable property since isomorphic graphs will be assigned codewords with the same length. Note that this is not sufficient to guarantee that all isomorphic graphs will be assigned *the same* codeword. Since this would imply a solution to GI, it remains an open problem. However, our graph encoding provides desirable tradeoffs between the expressivity of the probabilistic model, its number of parameters, and computational complexity, since it can adapt to different graph distributions using only a few parameters and solving GI only for small graphs.

### 4.4 Selecting a hypothesis by minimising the total description length

Putting everything together, in order to encode a graph dataset $\mathcal{G}$ sampled i.i.d. from $\mathfrak{G}$ we minimise the total description length

$$\min_{\theta, D, \phi} \sum_{G \in \mathcal{G}} \mathrm{L}_\phi(\mathsf{PART}_\theta(G) \mid D) + \mathrm{L}(D) \qquad (10)$$

with respect to the parameters $\theta$ of the parametric partitioning algorithm, the dictionary $D$, and the parameters $\phi$ of the probabilistic model. Eq. (10) is a typical two-part Minimum Description Length (MDL) objective [92, 90]. Using standard MDL terminology, the tuple $(\theta, D, \phi)$ is a *point hypotheses* and minimising (10) amounts to finding the simplest hypothesis that best describes the data.

## 5 Theoretical analysis: quadratic and linear gains

The following section performs a comparative analysis of the description length growth rate of various graph compressors. We compare PnC against two strong baselines: (a) The code length $L_{part}$ induced by a pure partitioning-based graph encoding. Here, a graph is decomposed into subgraphs and cuts, but the distribution of subgraphs is not modelled (i.e., both subgraphs and cuts are encoded with a null model $L_{part}(G) = L_{null}(\mathcal{H}) + L_{null}(C|\mathcal{H})$). (b) The code length of encodings that do not rely on partitioning but encode each graph as a whole. Importantly, our results hold even for those baselines that encode the isomorphism class of each graph, rather than the graph itself, such as the Erdős-Renyi model for unlabelled graphs of $n$ vertices: $L_{ER-S}(G) = \log|\mathfrak{G}_{n,m}| + \log(n^2 + 1)$, where $\mathfrak{G}_{n,m}$ is the set of all graphs with $n$ vertices and $m$ edges. $L_{ER-S}(G)$ serves as lower bound to typical encodings such as that of Eq. (2), but can be impractical to implement due to the complexity of GI. The analysis of additional baselines can be found in Appendix A.

Our main theorem shows that, under mild conditions on the underlying graph distribution, the expected description lengths of the compared encodings are totally ordered:

**Theorem 1.** *Consider a distribution $p$ over graphs with $n$ vertices and a partitioning algorithm that decomposes a graph into $b$ blocks of $k = O(1)$ vertices. Then it holds that:*

$$\mathbb{E}_{G\sim p}[L_{PnC}(G)] \overset{(1b)}{\lesssim} \mathbb{E}_{G\sim p}[L_{part}(G)] \overset{(1a)}{\lesssim} \mathbb{E}_{G\sim p}[L_{ER-S}(G)] \tag{11}$$

*under the following conditions:*

*(1a)* $\frac{\log(k^2+1)}{k^2}) + \bar{H}_{m_{ij}} < \bar{H}_m$, *where* $\bar{H}_{m_{ij}} = \mathbb{E}_{G\sim p}[H(\frac{m_{ij}}{k^2})]$ *and* $\bar{H}_m = \mathbb{E}_{G\sim p}[H(\frac{m}{n^2})]$ *is the expected binary entropy of the cut size $m_{ij}$ between two subgraphs and that of the total number of edges $m$, respectively.*

*(1b)* $|D| < (k^2+1)2^{k^2\bar{H}_{m_i}}$, *where $|D|$ is the size of the dictionary and* $\bar{H}_{m_i} = \mathbb{E}_{G\sim p}[H(\frac{m_i}{k^2})]$ *the expected binary entropy of the number of edges $m_i$ in a subgraph.*

*The compression gains are:*

$$\mathbb{E}_{G\sim p}[L_{Part}(G)] \lesssim \mathbb{E}_{G\sim p}[L_{ER-S}(G)] - n^2\Big(\bar{H}_m - \frac{\log(k^2+1)}{k^2} - \bar{H}_{m_{ij}}\Big) \tag{12}$$

and

$$\mathbb{E}_{G\sim p}[L_{PnC}(G)] \lesssim \mathbb{E}_{G\sim p}[L_{part}(G)] - nk(1-\delta)\Big(\bar{H}_{m_i} - \frac{\mathbb{H}(D) - \log(k^2+1)}{k^2}\Big), \tag{13}$$

*where $1-\delta$ is the probablity that a subgraph belongs in the dictionary and $\mathbb{H}(D) = \mathbb{H}_{a\sim q_\phi(a)}[a]$ is the entropy of the distribution on dictionary atoms $q_\phi(a)$.*

Theorem 1 provides insights on the compressibility of certain graph distributions given their structural characteristics. In particular, we can make the following remarks: (a) Condition (1a) can be satisfied even for very small values of $k$ as long as the graphs possess community structure. Perhaps counter-intuitively, when $k = O(1)$ we can satisfy the condition even if the communities have $O(n)$ size by splitting them into smaller subgraphs. This is possible because, in contrast to the majority of graph partitioning objectives that are based on minimum cuts, the compression objective attains its minimum when the cuts have "low entropy". Since communities that are tightly internally connected have large cuts, $\bar{H}_{m_{ij}}$ and the code length will be kept small. This is a key observation that strongly motivates the use of partitioning for graph compression. (b) Condition (1b) provides an upper bound to the size of the dictionary, which can be easily satisfied for moderately small values of $k$. More importantly, the dependence of the compression gain on the entropy $\mathbb{H}(D)$, reveals that dictionary atoms should be frequent subgraphs in the distribution, confirming our intuition. The bounds also show that, since the probabilities of the atoms are estimated from the data, PnC does not need to make assumptions about the inner structure of the subgraphs and can adapt to general distributions.

In Appendix A.4, we also provide theoretical evidence on the importance of of encoding dictionary subgraphs as isomorphism classes instead of adjacency matrices, which is related to challenge C1 mentioned in the introduction. In particular, Theorem 2 shows that, if isomorphism is not taken into account, the number of bits that will be lost will grow linearly with the number of vertices. All proofs and detailed assumptions can be found in Appendix A.

## 6 Optimisation and learning algorithms

We turn our focus to learning algorithms for the optimisation of the MDL objective (10). The following sections explain how each parametric component of PnC is learned.

**Subgraph encoding $\phi$.** The graph encoding is parametrised as follows: $q_\phi(a_i)$ and $q_\phi(b)$ are parametrised by learnable variables that are converted into categorical distributions over the dictionary atoms and the number of vertices respectively, using a softmax function. Similarly, $\delta_\phi$ is parametrised by a learnable variable converted to a probability via the sigmoid function.

**Dictionary $D$.** Let $\mathfrak{U} = \{a_1, a_2, \ldots a_{|\mathfrak{U}|}\}$ be a practically enumerable universe and define $\mathbf{x} = (x_1, x_2, \ldots, x_{|\mathfrak{U}|})$ as

$$x_i = \begin{cases} 1 & \text{if } a_i \in D \\ 0 & \text{otherwise.} \end{cases}$$

Thus, $x_i$ indicates whether $D$ contains subgraph $a_i$. Now, optimising w.r.t the dictionary amounts to finding the binary assignments for $\mathbf{x}$ that minimise (10). To circumvent the combinatorial nature of this problem, we apply the continuous relaxation $\hat{x}_i = \sigma(\psi_i), \forall i \in \{0, 1, \ldots, |\mathfrak{U}|\}$, where $\sigma$ is the sigmoid, $\psi_i$ are learned continuous variables, and $\hat{x}_i \in [0, 1]$ a fractional alternative to $x_i$. Appendix B.3 shows how (10) can be re-written w.r.t. $\mathbf{x}$ and optimised by using the surrogate gradient w.r.t $\hat{x}_i$.

It is important to note that, in practice, we do not have to introduce indicator variables for the entire universe: Since most subgraphs $a_i$ will be never encountered in the graph distribution, we build the universe adaptively during training, by progressively adding the different graphs that the partitioning algorithm yields. We also allow the universe to contain subgraphs of size up to $k = O(1)$, in order to ensure that the isomorphism testing between atoms and subgraphs can be efficiently computed.

**Parametric graph partitioning algorithm.** Finding the graph partitions that minimise (10) in principle requires searching in the space of partitioning algorithms. Instead, we constrain this space via a differentiable parametrisation that allows us to perform gradient-based optimisation. Currently, learning to partition is an open problem, as to the extent of our knowledge known neural approaches require a fixed number of clusters [93–95] or do not guarantee that the subgraphs are connected [96].

Our *Neural Partitioning* is a randomised algorithm parametrised with a graph neural network (GNN). When run on a graph, the GNN outputs a random $(\mathcal{H}, C)$ together with a corresponding probability $p_\theta^{\text{GNN}}(\mathcal{H}, C|G)$ and training is performed by estimating the gradients w.t.t. $\theta$ with REINFORCE [97]. Our algorithm proceeds by iteratively sampling (and removing) subgraphs from the graph until it becomes empty. At each step $t$ we select a subgraph $H_t$, by first sampling its vertex count $k_t$, and subsequently sampling at most $k_t$ vertices. To guarantee connectivity, we also sample the vertices iteratively and mask-out the probabilities outside the pre-selected vertices' neighbourhoods. The complexity of the algorithm is $O(n)$, where $n$ the number of the vertices of the graph. Please refer to Appendix B.4 for an in-depth explanation of the algorithm and relevant implementation details. We stress that we mainly consider this algorithm as a proof of concept that we ablate against other non-parametric partitioning algorithms. A plethora of solutions can be explored in a parametric setting and we welcome future work in this direction.

**MDL objective.** Given dataset $\mathcal{G}$, we train all components by minimising the description length:

$$L(\mathcal{G}) = \mathrm{L}_{\mathbf{x}}(D) + \sum_{G \in \mathcal{G}} \mathbb{E}_{(\mathcal{H}, C) \sim p_\theta^{\text{GNN}}(\mathcal{H}, C|G)}[\mathrm{L}_{\phi, \mathbf{x}}(\mathcal{H}, C|D)]. \tag{14}$$

Taking the expectation over the GNN output $(\mathcal{H}, C) \sim p_\theta^{\text{GNN}}(\mathcal{H}, C|G))$, we calculate the gradients as: $\nabla_\phi L(\mathcal{G}) = \sum_{G \in \mathcal{G}} \mathbb{E}[\nabla_\phi \mathrm{L}_{\phi, \mathbf{x}}(\mathcal{H}, C)|D)]$, $\nabla_{\hat{\mathbf{x}}} L(\mathcal{G}) = \nabla_{\hat{\mathbf{x}}} \mathrm{L}_{\hat{\mathbf{x}}}(D) + \sum_{G \in \mathcal{G}} \mathbb{E}[\nabla_{\hat{\mathbf{x}}} \mathrm{L}_{\phi, \hat{\mathbf{x}}}(\mathcal{H}, C|D)]$, and $\nabla_\theta L(\mathcal{G}) = \sum_{G \in \mathcal{G}} \mathbb{E}[\mathrm{L}_{\phi, \mathbf{x}}(\mathcal{H}, C|D) \nabla_\theta \ln p_\theta^{\text{GNN}}(\mathcal{H}, C|G)]$.

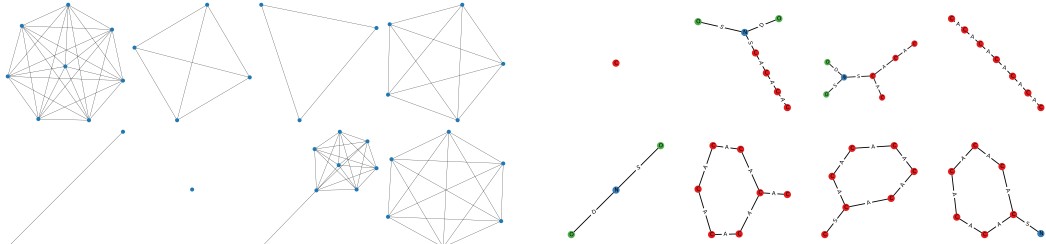

Figure 2: PnC + Neural Part. - Most probable graphs in the IMDB-B dataset (left) and the attributed MUTAG dataset (right). Atom types and bond types are represented as vertex and edge attributes.

## 7    Empirical Results

We evaluate our framework in a variety of datasets: small molecules, proteins and social networks [98–102]. Across all methods, we assume an optimal encoder that attains the entropy lower bound (this is often realistic since modern entropy coders asymptotically approach it) to evaluate the expressive power of each model independently of the encoder. We measure the description length of the data as their negative log-likelihood (NLL) under each probabilistic model, as well as the total description length by adding the cost of the parameters which need to be transmitted to the decoder (see Appendix D for details).

**Baselines.** We aim to assess representative approaches across the entire spectrum of graph probabilistic models, i.e., from completely uninformative non-parametric distributions to overparametrised neural generative models. We consider the following types of compressors: (a) *Null models.* We select the *uniform* model, where all edges are assumed to be sampled independently with probability equal to 0.5, the *edge list* model, a typical graph representation, and the *Erdős-Renyi* model (Eq. (2)). (b) *Partitioning-based.* non-parametric methods that aim at grouping vertices in tightly-connected clusters. They can be used for any type of sparse matrix [103] and are based on the assumption that there exists a hidden community structure in the graph. The partioning algorithms used are *SBM fitting* [44–47], Louvain [104] and Label Propagation [105] clustering. The encoding we use to encode the clusters corresponds exactly to the SBM assumptions, hence the partitioning-based results are always superior for this approach. (c) *Likelihood-based neural compressors.* As with any likelihood-based model, graph generative models can be transformed into graph compressors. We evaluate the original GraphRNN [66] and GRAN [68] networks, as well as smaller instantiations that have undergone model compression using the Lottery Ticket Hypothesis algorithm [106].

**Results.** Tables 1 and 2 report the compression quality of each method measured in terms of the average number of bits required to store each edge in a dataset (bpe). We present four variants of PnC, differing on the type of partitioning algorithm used [105, 104, 47]. We report separately the cost of compressing the data as well as the total cost (including the parameters). Several observations can be made with regards to the baselines:

First off, *off-the-shelf likelihood-based neural approaches are poor compressors due to failing to address challenge C3.* These models exhibit an unfavorable trade-off between the data and model complexity, often requiring significantly more bpe than the null models. Although model compression techniques can alleviate this tradeoff (especially for larger datasets, e.g., pruned GraphRNN on ZINC), in most of the cases the compression ratios required to outperform PnC are significantly higher than the best that have been reported in the literature (See Table 4 in the Appendix). Perhaps more importantly, it is unclear how to optimise the model description length during training (one of the few exceptions is [107]) and usually model compression might be tedious and is based on heuristics [108–111]. See Appendix C.2 for more details and additional experiments.

In addition, as expected from Theorem (1a), *non-parametric clustering algorithms work well when the dataset has a strong community structure, but are not a good choice for more structured datasets.* For instance, the best clustering algorithm requires 2.4× more bpe for ZINC than the best PnC. *PnC variants achieve the best compression in all datasets considered.* This follows from Theorem (1b), since the learned dictionaries are relatively small, and confirms our hypothesis that our framework is sufficiently flexible to account for the particularities of each dataset. As seen, neural partitioning

Table 1: Average bits per edge (bpe) for molecular graph datasets. **First**, **Second**, **Third**

| Method type | Graph type | Small Molecules | | | | | | | | |
| | Dataset name | MUTAG | | | PTC | | | ZINC | | |
| | | data | total | params | data | total | params | data | total | params |
| Null | Uniform (raw adjac.) | - | 8.44 | - | - | 17.43 | - | - | 10.90 | - |
| | Edge list | - | 7.97 | - | - | 9.38 | - | - | 8.60 | - |
| | Erdős-Renyi | - | 4.78 | - | - | 5.67 | - | - | 5.15 | - |
| Partitioning (non-parametric) | SBM-Bayes | - | 4.62 | - | - | 5.12 | - | - | 4.75 | - |
| | Louvain | - | 4.80 | - | - | 5.27 | - | - | 4.77 | - |
| | PropClust | - | 4.92 | - | - | 5.40 | - | - | 4.85 | - |
| Neural (likelihood) | GraphRNN | 1.33 | 3338.21 | 388K | 1.57 | 1394.59 | 389K | 1.62 | 43,16 | 388K |
| | GRAN | 0.81 | 12557.75 | 1460K | 2.18 | 5269.82 | 1470K | 1.30 | 157.7 | 1461K |
| | GraphRNN (pruned) | 1.95 | 12.39 | 1.08K | 2.16 | 6.71 | 1.10K | 1.79 | **2.02** | 1.90K |
| | GRAN (pruned) | 2.59 | 24.56 | 2.23K | 4.31 | 14.00 | 2.36K | 3.26 | 3.47 | 1.69K |
| PnC | PnC + SBM | 3.81 | 4.11 | 49 | 4.38 | 4.79 | 155 | 3.34 | 3.45 | 594 |
| | PnC + Louvain | 2.20 | **2.51** | 47 | 2.68 | **3.14** | 166 | 1.96 | **1.99** | 196 |
| | PnC + PropClust | 2.42 | **3.03** | 63 | 3.38 | **4.02** | 178 | 2.20 | 2.35 | 726 |
| | PnC + Neural Part. | 2.17±0.02 | **2.45±0.02** | 46±1 | 2.63±0.26 | **2.97±0.14** | 143±31 | 2.01±0.02 | **2.07±0.03** | 384±105 |

Table 2: Average bits per edge (bpe) for social and protein graph datasets. **First**, **Second**, **Third**

| Method type | Graph type | Biology | | | Social Networks | | | | | |
| | Dataset name | PROTEINS | | | IMDB-B | | | IMDB-M | | |
| | | data | total | params | data | total | params | data | total | params |
| Null | Uniform (raw adjac.) | - | 24.71 | - | - | 2.52 | - | - | 1.83 | - |
| | Edge list | - | 10.92 | - | - | 8.29 | - | - | 7.74 | - |
| | Erdős-Renyi | - | 5.46 | - | - | 1.94 | - | - | 1.32 | - |
| Partitioning (non-parametric) | SBM-Bayes | - | 3.98 | - | - | **0.80** | - | - | **0.60** | - |
| | Louvain | - | 3.95 | - | - | 1.22 | - | - | 0.88 | - |
| | PropClust | - | 4.11 | - | - | 1.99 | - | - | 1.37 | - |
| Neural (likelihood) | GraphRNN | 2.03 | 156.99 | 392K | 1.03 | 132.27 | 395K | 0.72 | 127.84 | 392K |
| | GRAN | 1.51 | 607.96 | 1545K | 0.26 | 488.88 | 1473K | 0.17 | 475.13 | 1467K |
| | GraphRNN (pruned) | 2.63 | 3.76 | 2.56K | 1.43 | 1.92 | 1.28K | 0.91 | 1.39 | 1.28k |
| | GRAN (pruned) | 4.28 | 5.11 | 1.78K | 0.84 | 1.75 | 2.38K | 0.55 | 1.41 | 2.31K |
| PnC | PnC + SBM | 3.26 | **3.60** | 896 | 0.50 | **0.54** | 198 | 0.38 | **0.38** | 157 |
| | PnC + Louvain | 3.34 | **3.58** | 854 | 0.96 | **1.02** | 202 | 0.66 | **0.70** | 141 |
| | PnC + PropClust | 3.42 | 3.68 | 866 | 1.45 | 1.64 | 241 | 0.93 | 1.04 | 178 |
| | PnC + Neural Part. | 3.34±0.25 | **3.51±0.23** | 717±61 | 1.00±0.04 | 1.05±0.04 | 186±25 | 0.66±0.05 | 0.72±0.05 | 178±14 |

performs in every case better, or on par with the combination of PnC with the Louvain algorithm. However, in the social network datasets, the combination of PnC with SBM achieves the best performance. This occurs because these networks fit nicely with the SBM inductive bias (which as a matter of fact is exactly that of low-entropy cuts), and most importantly, due to the fact that the clusters recovered by the SBM are small and repetitive, which makes them ideal for the PnC framework. We also observe that there is room for improvement for the neural partitioning variant, and hypothesise that a more powerful parametrised algorithm can be designed. Since learnable partitioning is still an open problem, we leave this research direction to future work.

Fig. 2 shows the most likely dictionary atoms for the IMDB-B and the MUTAG dataset (also including attributes - Appendix C.3 provides additional experiments). Observe that cliques or near-cliques and typical molecular substructures, such as carbon cycles and junctions are recovered for social networks and molecules respectively. This clearly highlights the connection between compression and pattern mining and provides evidence for potential applications of our framework.

# 8   Conclusion

This paper marks an important step towards learnable entropy-based graph compression. To the best of our knowledge, our work represents the first attempt to address the basic principles of parametric compressors of unlabelled graphs learned from observations. In addition, we suggest practical instantiations of our framework that can be trained with gradient-based optimisation, accounting for the total description length, hence aiming for the largest possible parsimony. A number of new research questions arise, such as how to design more expressive, albeit parsimonious distribution estimators, how to improve the neural partitioning algorithms, and how to ensure scalability to single large networks that pose a significant challenge w.r.t the memory constraints of GPUs. We hope that our work will inspire further research in this emerging research area.

## Acknowledgements and Disclosure of Funding

We would like to thank the anonymous reviewers for their valuable feedback and suggestions to improve our paper. This project was partially funded by the ERC Consolidator Grant No. 724228 - LEMAN. Giorgos Bouritsas is partially supported by a PhD scholarship from the Dpt. of Computing, Imperial College London. Andreas Loukas and Nikolaos Karalias thank the Swiss National Science Foundation for supporting them in the context of the project "Deep Learning for Graph Structured Data", grant number PZ00P2 179981. Michael Bronstein acknowledges support from Google Faculty awards and the Royal Society Wolfson Research Merit award.

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
