## A  Theoretical analysis

### A.1  Preliminaries

Recall the definition of the binary entropy $H(p) = -p \log p - (1-p) \log(1-p)$. A useful approximation that will be used in the analysis is the following:

$$\log \binom{n}{m} \approx n \, H\left(\frac{m}{n}\right), \tag{1}$$

which holds when both $n$ and $m$ grow at the same rate, i.e., $m \lesssim n$, and can be derived using Stirling's approximation.

For the following comparisons, we will be considering a graph distribution with $n$ vertices. For better exposition, the analysis will be performed for directed graphs. The same trends in the bounds hold also for undirected graphs.

**Conventional graph encodings.** We consider two types of baseline graph encodings:

- Uniform: $L_{\text{unif-G}}(G) = n^2$. Here no assumptions are made about the graph; all graphs are considered to be equally probable.

- Erdős-Renyi: $L_{\text{ER-G}}(G) = \log \binom{n^2}{m} + \log(n^2 + 1) \approx n^2 H_m + \log(n^2 + 1)$, where $m$ the number of edges and $H_m = H\left(\frac{m}{n^2}\right)$. This baseline is efficient at encoding graphs that are either very sparse or very dense.

Though the above encodings assign the same probability to isomorphic graphs, they map them to different codewords. Hence, they are redundant when dealing with unlabelled graphs. The following variants are more efficient by taking into account isomorphism:

- Uniform - Isomorphism classes: $L_{\text{unif-S}}(G) = \log |\mathfrak{G}_n| \approx n^2 - n \log n$.

- Erdős-Renyi - Isomorphism classes: $L_{\text{ER-S}}(G) = \log |\mathfrak{G}_{n,m}| + \log(n^2 + 1) \approx n^2 H_m + \log(n^2 + 1) - n \log n$,

where $\mathfrak{G}_n$ and $\mathfrak{G}_{n,m}$ are the set of all graphs with $n$ vertices and the set of all graphs with n vertices and $m$ edges, respectively. In both cases, we used the fact that asymptotically almost all graphs are rigid i.e., that they have only the trivial automorphism [1].

Observing that all four encodings asymptotically grow quadratically with the number of nodes we can derive the following lemma:

**Lemma 1.** *Consider a graph distribution $p$ over graphs with $n$ vertices and denote by $\bar{H}_m = \mathbb{E}_{G \sim p}[H(\frac{m}{n^2})]$ the expected value of the binary entropy of the number of edges $m$. If $\bar{H}_m < 1$, then the expected description lengths of the baseline models are asymptotically ordered as follows:*

$$\mathbb{E}_{G \sim p}[L_{\text{ER-S}}(G)] \lesssim \mathbb{E}_{G \sim p}[L_{\text{ER-G}}(G)] \lesssim \mathbb{E}_{G \sim p}[L_{\text{unif-S}}(G)] \lesssim \mathbb{E}_{G \sim p}[L_{\text{unif-G}}(G)] \tag{2}$$

*The compression gain when encoding isomorphism classes instead of labelled graphs is $\Theta(n \log n)$, while that of the Erdős-Renyi encoding compared to the uniform one is $\Theta\left(n^2(1 - \bar{H}_m)\right)$.*

The proof follows directly from the equations above. Since most real-world graphs are sparse, then the condition $\bar{H}_m < 1$ is almost always true.

**Partitioning (only).** The partitioning models considered in this analysis assume that each graph is clustered into $b$ subgraphs of $k$ vertices each (i.e., $b = \frac{n}{k}$) and that the intra- and inter-subgraph edges are encoded independently[1]. Note that the preamble terms that encode the number of blocks in the partition and the number of vertices per block are unnecessary since $k$ is fixed. Overall, the code length is given by

$$L_{\text{Part}}(G) = L(\mathcal{H}) + L(C|\mathcal{H}),$$

---

[1]To make the analysis more tanglible, we examine here a slightly simpler encoding than the one used in the experiments - see Eq. (9).

with the two terms defined respectively as follows:

$$\mathrm{L}(\mathcal{H}) = \sum_{i=1}^{b} \left( \log(k^2 + 1) + \log \binom{k^2}{m_i} \right) \quad \text{and} \quad \mathrm{L}(C|\mathcal{H}) = \sum_{i \neq j}^{b} \left( \log(k^2 + 1) + \log \binom{k^2}{m_{ij}} \right). \tag{3}$$

Above, $m_i$ is the number of edges in subgraph $H_i$ and $m_{ij}$ is the size of the cut between subgraphs $H_i$ and $H_j$. In both cases, the first term encodes the number of edges, and the second their arrangement across the vertices.

**Partition and Code (PnC).** We make the same assumptions for PnC as in $\mathrm{L}_{\mathrm{Part}}(G)$: the graph is partitioned into $b$ subgraphs of $k$ vertices and the same encoding of non-dictionary subgraphs $\mathcal{H}$ and cuts $C$ is used. The number of dictionary subgraphs $b_{\mathrm{dict}}$ is encoded with a Binomial distribution and the dictionary subgraphs $\mathcal{H}_{\mathrm{dict}}$ themselves with a Multinomial as in Eq. (6) and (7) of the main paper. The overall encoding length is

$$\mathrm{L}_{\mathrm{PnC}}(G|D) = \mathrm{L}(b_{\mathrm{dict}}) + \mathrm{L}(\mathcal{H}_{\mathrm{dict}}|b_{\mathrm{dict}}, D) + \mathrm{L}(\mathcal{H}_{\mathrm{null}}|b_{\mathrm{null}}, D) + \mathrm{L}(C|\mathcal{H}), \tag{4}$$

Each dictionary atom is encoded using any of the null models mentioned above, hence $\mathrm{L}(D) = O(|D|k^2)$.

## A.2   Why partitioning? Partitioning vs Null Models

As a warm-up, we will discuss the case of encodings based on pure graph partitioning, such as the Stochasic Block Model ("Partioning non-parametric" in the tables of the main paper). We remind the reader that these encodings do *not* take into account the isomorphism class of the identified subgraphs but rely on a null model to encode them.

In the following, we derive a sufficient condition for the sparsity of the connections between subgraphs, under which partitioning-based encodings will yield smaller expected description length than the baseline null models. Formally:

**Theorem 1a.** *Let every $G \sim p$ be partitioned into $b$ blocks of $k = O(1)$ vertices and suppose that the partitioning-based encoding of Eq. (3) is utilised. The following holds:*

$$\mathbb{E}_{G \sim p}[\mathrm{L}_{\mathrm{Part}}(G)] \lesssim \mathbb{E}_{G \sim p}[\mathrm{L}_{\mathrm{ER\text{-}S}}(G)] - n^2 \left( \bar{\mathrm{H}}_m - \frac{\log(k^2 + 1)}{k^2} - \bar{\mathrm{H}}_{m_{ij}} \right), \tag{5}$$

*where $\bar{\mathrm{H}}_{m_{ij}} = \mathbb{E}_{G \sim p}[\mathrm{H}\left(\frac{m_{ij}}{k^2}\right)]$ and $\bar{\mathrm{H}}_m = \mathbb{E}_{G \sim p}[\mathrm{H}\left(\frac{m}{n^2}\right)]$ are the expected binary entropy of the cuts and of the total number of edges, respectively.*

*Proof.*

$$\mathbb{E}_{G \sim p}[\mathrm{L}_{\mathrm{part}}(G)] \approx \mathbb{E}_{G \sim p} \Big[ \sum_{i=1}^{b} \left( \log(k^2 + 1) + k^2 \mathrm{H}\left(\frac{m_i}{k^2}\right) \right) + \sum_{i \neq j}^{b} \left( \log(k^2 + 1) + k^2 \mathrm{H}\left(\frac{m_{ij}}{k^2}\right) \right) \Big]$$

$$= \frac{n}{k} \left( \log(k^2 + 1) + k^2 \bar{\mathrm{H}}_{m_i} \right) + \left(\frac{n^2}{k^2} - \frac{n}{k}\right) \left( \log(k^2 + 1) + k^2 \bar{\mathrm{H}}_{m_{ij}} \right)$$

$$= n^2 \left( \frac{\log(k^2 + 1)}{k^2} + \bar{\mathrm{H}}_{m_{ij}} \right) + nk \left( \bar{\mathrm{H}}_{m_i} - \bar{\mathrm{H}}_{m_{ij}} \right)$$

$$= \mathbb{E}_{G \sim p}[\mathrm{L}_{\mathrm{ER\text{-}S}}(G)] - n^2 \left( \bar{\mathrm{H}}_m - \frac{\log(k^2 + 1)}{k^2} - \bar{\mathrm{H}}_{m_{ij}} \right)$$

$$+ nk \left( \bar{\mathrm{H}}_{m_i} - \bar{\mathrm{H}}_{m_{ij}} + \log n \right) - \log(n^2 + 1)$$

$$\lesssim \mathbb{E}_{G \sim p}[\mathrm{L}_{\mathrm{ER\text{-}S}}(G)] - n^2 \left( \bar{\mathrm{H}}_m - \frac{\log(k^2 + 1)}{k^2} - \bar{\mathrm{H}}_{m_{ij}} \right),$$

where we assumed that $m \lesssim n^2$, $m_i \lesssim k^2$, $m_{ij} \lesssim k^2$, and in the last step we derive an asymptotic inequality using the dominating quadratic term. In other words, partitioning-based encoding is

quadratically superior to the best null model whenever there exists a $k$ such that

$$\frac{\log(k^2+1)}{k^2} < \bar{\mathrm{H}}_m - \bar{\mathrm{H}}_{m_{ij}}.$$

The above concludes the proof. $\qquad\square$

## A.3 The importance of the dictionary: PnC vs Partitioning

We proceed to mathematically justify why encoding subgraphs with a dictionary (the "Code" part in PnC) can yield extra compression gains compared to pure partitioning-based encodings.

As our main theorem shows, utilising a dictionary allows us to reduce the linear $O(n)$ terms of the partitioning-based description length:

**Theorem 1b.** *Let every $G \sim p$ be partitioned into $b$ blocks of $k = O(1)$ vertices and suppose that the PnC encoding of Eq. (4) is used. If there exists a dictionary such that $|D| < (k^2+1)2^{k^2\bar{H}_{m_i}}$ with $\bar{H}_{m_i} = \mathbb{E}_{G\sim p}[H(\frac{m_i}{k^2})]$ being the expected binary entropy of the subgraph edges, then it holds that:*

$$\mathbb{E}_{G\sim p}[\mathrm{L}_{\mathrm{PnC}}(G)] \lesssim \mathbb{E}_{G\sim p}[\mathrm{L}_{\mathrm{part}}(G)] - nk(1-\delta)\left(\bar{\mathrm{H}}_{m_i} - \frac{\mathbb{H}(D) - \log(k^2+1)}{k^2}\right), \qquad (6)$$

*where $1 - \delta$ is the probability that a subgraph belongs in the dictionary and $\mathbb{H}(D)$ is the entropy of the distribution $q$ over the dictionary atoms.*

*Proof.* We will analyse the description length of each of the components of Eq. (4).

*Number of dictionary subgraphs.* The expected description length of the number of subgraphs is equal to the entropy of the binomial distribution:

$$\begin{aligned}
\mathbb{E}_{G\sim p}[\mathrm{L}(b_{\mathrm{dict}})] &= \frac{1}{2}\log\left(2\pi eb\delta(1-\delta)\right) + O\left(\frac{1}{b}\right) \\
&= \frac{1}{2}\log\left(2\pi e\delta(1-\delta)\right) + \frac{1}{2}\log\left(\frac{n}{k}\right) + O\left(\frac{k}{n}\right) = O\left(\log\frac{n}{k}\right)
\end{aligned}$$

*Dictionary subgraphs.* The expected description length of the subgraphs that belong in the dictionary amounts to the entropy of the multinomial distribution and can be upper bounded as follows:

$$\begin{aligned}
\mathbb{E}_{G\sim p}[\mathrm{L}(\mathcal{H}_{\mathrm{dict}}|b_{\mathrm{dict}}, D)] &= \mathbb{E}_{G\sim p}\left[-\log\frac{b_{\mathrm{dict}}!}{\prod_{a\in D}b_a!} - \sum_{a\in D}b_a\log q(a)\right] \\
&\leq \mathbb{E}_{G\sim p}[-\sum_{a\in D}b_a\log q(a)] \\
&= -\sum_{a\in D}\mathbb{E}_{G\sim p}[b_a]\log q(a) \\
&= -b(1-\delta)\sum_{a\in D}q(a)\log q(a) = \frac{n}{k}(1-\delta)\mathbb{H}(D) \leq \frac{n}{k}(1-\delta)\log|D|,
\end{aligned}$$

where we used the fact that $b_{\mathrm{dict}}! \geq \prod_{a\in D}b_a!$. The term $\mathbb{H}(D) = \mathbb{H}_{a\sim q(a)}[a] = -\sum_{a\in D}q(a)\log q(a)$ corresponds to the entropy of the dictionary distribution.

*Non-dictionary subgraphs.* Assuming that the subgraph edges $m_i$ are independent from the number of non-dictionary subgraphs $b - b_{\mathrm{dict}}$, then the expected description length of the non-dictionary subgraphs becomes:

$$\mathbb{E}_{G\sim p}[\mathrm{L}(\mathcal{H}_{\mathrm{null}}|b_{\mathrm{null}}, D)] \approx \mathbb{E}_{G\sim p}\left[\sum_{i=1}^{b-b_{\mathrm{null}}}\left(\log(k^2+1) + k^2\mathrm{H}_{m_i}\right)\right] = \frac{n}{k}\delta\left(\log(k^2+1) + k^2\bar{\mathrm{H}}_{m_i}\right)$$

Overall, using the same derivation for the cuts as in Theorem 1a, we obtain:

$$\mathbb{E}_{G\sim p}[\mathrm{L_{PnC}}(G|D)] \lesssim O(\log\frac{n}{k}) + n^2\left(\frac{\log(k^2+1)}{k^2} + \bar{\mathrm{H}}_{m_{ij}}\right)$$

$$+ n\left(\delta k\bar{\mathrm{H}}_{m_i} + (1-\delta)\left(\frac{\mathbb{H}(D) - \log(k^2+1)}{k}\right) - k\bar{\mathrm{H}}_{m_{ij}}\right)$$

$$= \mathbb{E}_{G\sim p}[\mathrm{L_{part}}(G)] + n(1-\delta)\left(\frac{\mathbb{H}(D) - \log(k^2+1)}{k} - k\bar{\mathrm{H}}_{m_i}\right) + O(\log\frac{n}{k})$$

Including the description length of the dictionary and amortising it over each graph in a dataset of $\mathcal{G}$ graphs, we conclude

$$\mathbb{E}_{G\sim p}[\mathrm{L_{PnC}}(G)] = \mathbb{E}_{G\sim p}[\mathrm{L_{part}}(G)] - nk(1-\delta)\left(\bar{\mathrm{H}}_{m_i} - \frac{\mathbb{H}(D) - \log(k^2+1)}{k^2}\right) + O(\log\frac{n}{k} + \frac{|D|}{|\mathcal{G}|}k^2).$$

Hence, if $k = O(1)$ and $|D| \ll |\mathcal{G}|$ (more precisely, the ratio $\frac{|D|}{|\mathcal{G}|}$ shouldn't grow with $n$), then a linear compression gain is obtained if:

$$\bar{\mathrm{H}}_{m_i} - \frac{\mathbb{H}(D) - \log(k^2+1)}{k^2} > 0 \Leftrightarrow \mathbb{H}(D) < \log(k^2+1) + k^2\bar{\mathrm{H}}_{m_i}.$$

The proof concludes by noting that the above condition is implied by $|D| < (k^2+1)2^{k^2\bar{\mathrm{H}}_{m_i}}$. $\qquad\square$

### A.4 The importance of subgraph isomorphism - Theorem 2

**Theorem 2.** *Let $p$ be a graph distribution that is invariant to isomorphism, i.e., $p(G') = p(G)$ if $G \cong G$. Moreover, consider any algorithm that partitions $G$ in $b$ subgraphs of $k$ vertices. Denote by $\mathrm{L_{PnC\text{-}G}}$ and $\mathrm{L_{PnC\text{-}S}}$ the description length of a PnC compressor that uses a dictionary of atoms encoded as labelled graphs and as isomorphism classes, respectively. The following holds:*

$$\mathbb{E}_{G\sim p}[\mathrm{L_{PnC\text{-}S}}(G)] \approx \mathbb{E}_{G\sim p}[\mathrm{L_{PnC\text{-}G}}(G)] - n(1-\delta)\log k \qquad (7)$$

*under the condition that almost all graphs in the dictionary are rigid[2].*

Importantly, the compression gains implied by the theorem hold independently of the size of the dictionary, applying e.g., also when the dictionary is equal to the universe and contains all graphs of size $k$ (which amounts to the traditional partitioning baselines).

*Proof.* Let $D_G$ be a dictionary of *labelled* graphs of $k$ vertices, i.e. graphs whose vertices are ordered, and $D_S$ the corresponding dictionary of *unlabelled* graphs, i.e., where the atoms are isomorphism classes.

In the context of this comparison we are only interested in the description length of the dictionary subgraphs. For simplicity we will assume that these are encoded with a *categorical* distribution instead of multinomial:

$$\mathbb{E}_{G\sim p}[\mathrm{L}(\mathcal{H}_{\mathrm{dict}}|b_{\mathrm{dict}}, D)] = \mathbb{E}_{G\sim p}\left[-\sum_{a\in D} b_a \log q(a)\right] = -b(1-\delta)\sum_{a\in D} q(a)\log q(a) = \frac{n}{k}(1-\delta)\mathbb{H}(D)$$

Hence, in order to compare the two variants, we are interested in the entropy $\mathbb{H}(D)$, which requires enumerating the possible outcomes of the categorical distribution, i.e., the dictionary atoms.

Denote with $S_a$ the isomorphism class of an atom $a$, i.e., $S_a = \{a' \in \mathfrak{G}_k | a \cong a'\}$ It is known that the size of each $S_a$ is $|S_a| = \frac{k!}{|\mathrm{Aut}(a)|}$ [2], where $\mathrm{Aut}(a)$ the automorphism group of $a$, i.e., all isomorphisms that map the adjacency matrix onto itself.

Since $p(G)$ is isomorphism invariant, then the same will hold for the subgraphs $H \subseteq G$, i.e. $p(H') = p(H)$ if $H \cong H$. Hence, regarding PnC-G, it should hold that for each atom $a$ in $D_G$, then all $a' \cong a$ should be also contained in the dictionary and assigned the same probability, i.e.,

---

[2]A rigid graph has only the trivial automorphism.

$q_G(a) = q_G(a')$. Therefore, the corresponding probabilties of isomorphism classes in the context of PnC-S should be as follows: $q_S(S_a) = \sum_{a \in S_a} q_G(a) = |S_a| q_G(a)$.

Then, using a similar argument to [3], we can derive the following for the entropy $\mathbb{H}_G(D)$:

$$\mathbb{H}_G(D) = - \sum_{a \in D_G} q_G(a) \log q(a)$$

$$= - \sum_{a \in D_G} \frac{q_S(S_a)}{|S_a|} \log \frac{q_S(S_a)}{|S_a|}$$

$$= - \sum_{S_a \in D_S} \sum_{a \in S_a} \frac{q_S(S_a)}{|S_a|} \log \frac{q_S(S_a)}{|S_a|}$$

$$= - \sum_{S_a \in D_S} q_S(S_a) \log \frac{q_S(S_a)}{|S_a|}$$

$$= \mathbb{H}_S(D) + \sum_{S_a \in D_S} q_S(S_a) \log |S_a| = \mathbb{H}_S(D) + \log k! - \sum_{S_a \in D_S} q_S(S_a) \log |\text{Aut}(a)|$$

At this point we will assume that almost all graphs in the dictionary are rigid, or more precisely we require that $\sum_{S_a \in D_S} q_S(S_a) \log |\text{Aut}(a)| \approx 0$, which can be also satisfied when non-rigid dictionary atoms have small probability. In practice, although for very small graphs of up to 4 or 5 vertices, many graphs have non-trivial automorphisms, this condition is easily satisfied for larger $k$ (but still of constant size w.r.t. $n$), that were also considered in practice. Then, the result immediately follows:

$$\mathbb{E}_{G \sim p}[L_{\text{PnC-G}}(\mathcal{H}_{\text{dict}}|b_{\text{dict}}, D)] \approx \mathbb{E}_{G \sim p}[L_{\text{PnC-S}}(\mathcal{H}_{\text{dict}}|b_{\text{dict}}, D)] + \frac{n}{k}(1 - \delta) \log k! \implies$$

$$\mathbb{E}_{G \sim p}[L_{\text{PnC-S}}(G)] \approx \mathbb{E}_{G \sim p}[L_{\text{PnC-G}}(G)] - n(1 - \delta) \log k,$$

where we used Stirling's approximation $\log k! \approx k \log k$. □

## B  Algorithmic Details

### B.1  Cut encoding

Denote the vertex count of subgraph $H_i$ as $k_i$. Further denote with $\boldsymbol{m}_c = \{m_{1,1}, m_{1,2}, \ldots, m_{b-1,b}\}$ the vector containing the number of edges between each subgraph pair $i, j$ and $m_c = \sum_{i<j}^{b} m_{ij}$. The $b$-partite graph $C$ containing the cuts will be encoded hierarchically, i.e. first we encode the total edge count $m_c$, then the pairwise counts $\boldsymbol{m}_c$ and finally, for each subgraph pair, we independently encode the arrangement of the edges. For each of these cases, a uniform encoding is chosen, following the rationale mentioned in Section 3 of the main paper. Hence, calculating the length of the encoding boils down to enumerating possible outcomes:

$$L(C|\mathcal{H}) = L(C, m_c, \boldsymbol{m}_c|\mathcal{H}) = L(m_c|\mathcal{H}) + L(\boldsymbol{m}_c|m_c, \mathcal{H}) + L(C|\boldsymbol{m}_c, m_c, \mathcal{H})$$

$$= \log\left(1 + \sum_{j>i}^{b} k_i k_j\right) + \log\binom{b(b-1)/2 + m_c - 1}{m_c} + \sum_{j>i}^{b} \log\binom{k_i k_j}{m_{ij}} \quad (8)$$

We make the following remarks: (a) The encoding is the same regardless of the isomorphism class of the subgraphs, and the only dependence arises from their number, as well as their vertex counts. (b) Small cuts are prioritised, thus the encoding has an inductive bias towards distinct clusters in the graph. (c) Our cut encoding bears resemblance to those used in non-parametric Bayesian inference for SBMs (e.g., see the section B.2 on the baselines and [4] for a detailed analysis of a variety of probabilistic models), although a central difference is that in these works the encodings also take the vertex ordering into account.

### B.2  Baseline Encodings

**Null models.** The description length of the uniform encoding is equal to $L(G) = \log(n_{max} + 1) + \binom{n}{2}$ and that of the edge list model is $L(G) = \log(n_{max} + 1) + \log\left(\binom{n}{2} + 1\right) + m \log \binom{n}{2}$.

**Clustering.** The encoding we used for the clustering baselines is optimal under SBM assumptions and is obtained from [4] with small modifications. It consists of the following uniform encodings: number of graph vertices, number of graph edges, number of blocks, number of vertices in each block, number of edges inside each block and between each pair of blocks, and finally the arrangements of intra- and inter-block edges (a detailed explaination for each term can be found in [4] and [5]):

$$
\mathrm{L}(G) = \log(n_{\max} + 1) + \log\left(n(n-1)/2 + 1\right) + \log(n) + \log\binom{n-1}{b-1}
$$

$$
+ \log\binom{b(b+1)/2 + m - 1}{m} + \sum_{i=1}^{b} \log\binom{\binom{k_i}{2}}{m_i} + \sum_{i<j}^{b} \log\binom{k_i k_j}{m_{ij}}
\tag{9}
$$

### B.3 Dictionary Learning - Continuous Relaxation

In the following section we will relax the Minimum Description Length objective (Eq. (10) in the main paper) by introducing the fractional membership variables $\hat{\mathbf{x}}$.

The dictionary description length, Eq. (5), can be trivially rewritten as follows:

$$
\mathrm{L}_{\hat{\mathbf{x}}}(D) = \sum_{a \in \mathfrak{U}} \hat{x}_a \, \mathrm{L}_{\mathrm{null}}(a).
\tag{10}
$$

Regarding the description length of the graphs, the membership variables are the ones that select when a subgraph is encoded as a dictionary atom or when with the help of the null model. Note that this is an additional explanation for the necessity of the the dual encoding: except for giving sufficient freedom to the partioning algorithm to choose non-dictionary atoms, it is also necessary for optimisation.

The relaxation of the graph description length was done as follows: $b_a(\hat{\mathbf{x}}) = \hat{x}_a b_a$, $b_{\mathrm{dict}}(\hat{\mathbf{x}}) = \sum_{a \in \mathfrak{U}} b_a(\hat{\mathbf{x}})$, and $q_{\phi,\hat{\mathbf{x}}}(a) = \frac{\hat{x}_a e^{\phi_a}}{\sum_{a' \in \mathfrak{U}} \hat{x}_{a'} e^{\phi_{a'}}}$. The rest of the components of the graph description length are unaffected from the choice of the dictionary. Now Eq. (6)-(8) can be rewritten as:

$$
\mathrm{L}_{\phi,\hat{\mathbf{x}}}(b_{\mathrm{dict}}, b_{\mathrm{null}}) = -\log\binom{b}{b_{\mathrm{dict}}(\hat{\mathbf{x}})} - b_{\mathrm{dict}}(\hat{\mathbf{x}}) \log(1 - \delta_\phi) - \left(b - b_{\mathrm{dict}}(\hat{\mathbf{x}})\right) \log(\delta_\phi) - \log q_\phi(b)
$$

$$
\mathrm{L}_{\phi,\hat{\mathbf{x}}}(\mathcal{H}_D | b_{\mathrm{dict}}, D) = -\log\left(b_{\mathrm{dict}}(\hat{\mathbf{x}})!\right) + \sum_{a \in \mathfrak{U}} \log\left(b_a(\hat{\mathbf{x}})!\right) - \sum_{a \in \mathfrak{U}} b_a(\hat{\mathbf{x}}) \log q_{\phi,\hat{\mathbf{x}}}(a)
$$

$$
\mathrm{L}_{\mathrm{null},\hat{\mathbf{x}}}(\mathcal{H}_{\mathrm{null}} | b_{\mathrm{null}}, D) = -\sum_{H \in \mathcal{H}} \log q_{\mathrm{null}}(H)(1 - \hat{x}_H), \text{ where } \hat{x}_H = \begin{cases} \hat{x}_i & \exists\, a_i \in \mathfrak{U} \text{ s.t. } H \cong a_i \\ 0 & \text{otherwise.} \end{cases}
\tag{11}
$$

To obtain a continuous version of the terms where factorials are involved we used the $\Gamma$ function, where $\Gamma(n+1) = n!$, for positive integers $n$. The rest of the terms are differentiable w.r.t $\hat{\mathbf{x}}$.

### B.4 Learning to Partition

We remind that our algorithm is based on a double iterative procedure: the external iteration refers to subgraph selection and the internal to vertex selection. In order for the algorithm to be able to make decisions, we maintain a representation of two states: the *subgraph state* $S_t^H = \{H_1, H_2, \dots H_t\}$ that summarises the decisions made at the subgraph level (external iteration) up to step $t$, and the *vertex state* $S_i^V = \{v_{t_1}, v_{t_2}, \dots v_{t_i}\}$ that summarises the decisions made at the vertex level (internal iteration) up to the i-th vertex selection. Overall, we need to calculate the probability of $S_T$, where $T$ is the number of iterations:

$$
p_\theta(S_T | G) = p_\theta(H_T | S_{T-1}^H, G) p_\theta(S_{T-1}^H | G) = \prod_{t=1}^{T} p_\theta(H_t | S_{t-1}^H, G)
$$

$$
= \prod_{t=1}^{T} \left( \prod_{i=1}^{k_t} p_\theta(v | S_{i-1}^V, k_t, S_{t-1}^H, G) \right) p_\theta(k_t | S_{t-1}^H, G)
\tag{12}
$$

Hence, the parametrisation of the algorithm boils down to defining the vertex count probability $p_\theta(k_t|S_{t-1}^H, G)$ and the vertex selection probability $p_\theta(v|S_{i-1}^V, k_t, S_{t-1}^H, G)$, where $v \in V_t$ and $V_t$ the set of the remaining vertices at the step $t$.

Now we explain in detail how we parametrise each term. First, we use a Graph Neural Network (GNN) to embed each vertex into a vector representation $\mathbf{h}(v) = \text{GNN}_v(G)$, while the graph itself is embedded in a similar way $\mathbf{h}(G) = \text{GNN}_G(G)$.

Each subgraph is represented by a permutation invariant function on the embeddings of its vertices, i.e., $\mathbf{h}(H_t) = \text{DeepSets}(\{\mathbf{h}(v)|v \in H_t\})$, where we used DeepSets [6] as a set function approximator. Similarly, the subgraph state summarises the subgraph representations in a permutation invariant manner to ensure that future decision of the algorithm do not depend on the order of the past ones: $\mathbf{h}(S_t^H) = \text{DeepSets}(\{\mathbf{h}(H_t)|H_t \in S_t^H\})$.

Given the above, the probability of the vertex count at step $t$ is calculated as follows:

$$p_\theta(k_t|S_{t-1}^H, G) = \text{softmax}_{k_t=1}^{|V_t|} \text{MLP}\big(\mathbf{h}(S_{t-1}^H), \mathbf{h}(G)\big), \tag{13}$$

where MLP is Multi-layer perceptron.

As mentioned above, the probability of the selection of each vertex is computed in a way that guarantees connectivity:

$$p_\theta(v|S_{i-1}^V, k_t, S_{t-1}^H, G) = \begin{cases} \text{softmax}_{v \in V_t}\text{MLP}\big(\mathbf{h}(S_{t-1}^H), \mathbf{h}(v)\big) & i = 0, v \in V_t \\ \text{softmax}_{v \in V_t \cap \mathcal{N}(S_{i-1}^V)}\text{MLP}\big(\mathbf{h}(S_{t-1}^H), \mathbf{h}(v)\big) & 0 < i < k_t, v \in V_t \cap \mathcal{N}(S_{i-1}^V) \\ 0 & \text{otherwise,} \end{cases} \tag{14}$$

where $\mathcal{N}(S_{i-1}^V)$ denotes the union of the neighbourhoods of the already selected vertices, excluding themselves: $\mathcal{N}(S_{i-1}^V) = \bigcup_{i'=1}^{i-1} \mathcal{N}(v_{t_{i'}}) - \{v_{t_1}, v_{t_2}, \ldots v_{t_{i-1}}\}$. Overall, the parameter set $\theta$ is the set of the parameters of the neural networks involved, i.e., GNNs, DeepSets and MLPs.

In the algorithm 1 we schematically illustrate the different steps described above.

---

**Algorithm 1:** Partitioning algorithm

---

**Input**: graph $G$
**Output**: partition $\mathcal{H}$
**Initialisations:** $\mathbf{h}(v) = \text{GNN}_v(G)$, $\mathbf{h}(G) = \text{GNN}_G(G)$, $V_1 = V$, $S_0^H = \emptyset$
$t \leftarrow 1$
**while** $V_t \neq \emptyset$ **do**
    $k_t \sim p_\theta(k_t|S_{t-1}^H, G)$ // `sample maximum vertex count`
    Initialise $S_0^V = \emptyset$
    **while** $i = 1 \leq k_t$ *and* $\mathcal{N}(S_{i-1}^V) \neq \emptyset$ **do**
        $v_{t_i} \sim p_\theta(v|S_{i-1}^V, k_t, S_{t-1}^H, G)$ // `sample new vertex`
        $S_i^V = S_{i-1}^V \cup \{v_{t_i}\}$
    **end**
    $H_t = S_i^V$
    $S_t^H = S_{t-1}^H \cup \{H_t\}$
    $t \leftarrow t + 1$
**end**
$\mathcal{H} = S_t^H$

---

**Limitations.** Below we list two limitations of the learnable partitioning algorithm that we would like to address in future work. First, it is well known that GNNs have limited expressivity which is bounded by the Weisfeiler Leman test [7, 8]. The most important implication of this is that they have difficulties in detecting and counting substructures [9]. Since in our case subgraph detection is crucial in order to be able to partition the graph into repetitive substructures, the expressivity of the GNN might be an issue. Although iterative sampling may mitigate this behaviour up to a certain extent, the GNN will not be able to express arbitrary randomised algorithms. Modern architectures such as [10, 11] might be more suitable for this task, which makes them good candidates for future exploration on the problem.

Second, more sophisticated inference schemes should be explored, since currently a partition is decoded from the randomised algorithm by taking a single sample from the learned distribution. In particular, currently at each step $t$ the algorithm can only sample $k_t$ vertices as dictated by the initial sampling on the vertex count. However, there might be benefit from expanding the subgraph more, or stopping earlier than $k_t$ when no other vertex addition can contribute towards a smaller description length. However, there is no control on the stopping criterion apart from the initial vertex count prediction. To this end, it is of interest to explore alternatives that will allow the algorithm to choose from a pool of candidate decisions based on the resulting description lengths (e.g., in hindsight). Further inspiration can be taken from a variety of clustering and graph partitioning algorithms, e.g. k-means or soft clustering in a latent space [12, 13], agglomerative [14, 15] and Markov Chain Monte Carlo as in [16] where a modified Metropolis-Hastings algorithm is proposed.

### B.5 Special cases of note

A pertinent question is whether one can determine the optimal way to partition a graph when minimising (10). Though a rigorous statement is beyond our current understanding, in the following we discuss two special cases:

*(a) Small predefined universe.* When the subgraphs are chosen from a small and predefined $\mathfrak{U}$, one may attempt to identify all the possible atom appearances in $G$ by repeatedly calling a subgraph isomorphism subroutine. The minimisation of (10) then simplifies to that of selecting a subset of subgraphs that have no common edges (as per the definition in Section 4.1). The latter problem can be cast as a discrete optimisation problem under an independent set constraint (by building an auxiliary graph the vertices of which are candidate subgraphs and two vertices are connected by an edge when two subgraphs overlap, and looking for an independent set that minimises the description length).

*(b) Unconstrained universe.* When $\mathfrak{U}$ contains all possible graphs, the problem can be seen as a special graph partitioning problem. However, contrary to traditional clustering algorithms [17, 18, 15, 14], our objective is not necessary optimised by finding small cuts between clusters (see Appendix A).

Since most independent set and partitioning problems are NP-hard, we suspect that similar arguments can be put forward for (10). This highlights the need to design learnable alternatives that can provide fast solutions without the need to be adapted to unseen data.

## C   Additional Experiments

### C.1   Ablation studies

**Compression of unseen data.** In the Tables 1 and 2 we report the compression rates (in bpe) of the training and the test data separately for all the PnC variants. As can be seen, in most of the cases the generalisation gap is small, which implies that there was no evidence of overfitting and the compressor can be used to unseen data with small degradation in the compression quality.

Table 1: Average negative log likelihood of train and test data in bpe (molecular distributions).

| Dataset name | MUTAG | | PTC | | ZINC | |
|---|---|---|---|---|---|---|
| Set | train | test | train | test | train | test |
| PnC + SBM | 3.81 | 3.85 | 4.40 | 4.25 | 3.33 | 3.41 |
| PnC + Louvain | 2.18 | 2.39 | 2.67 | 2.74 | 1.96 | 1.97 |
| PnC + PropClust | 2.37 | 2.89 | 3.33 | 3.83 | 2.19 | 2.27 |
| PnC + Neural Part | 2.16±0.02 | 2.28±0.03 | 2.64±0.24 | 2.59±0.21 | 2.01±0.02 | 2.03±0.01 |

**Out-of-distribution compression.** In the following experiment we tested the ability of PnC to compress data sampled from different distributions. In particular we trained the Neural Partitioning variant on one of the MUTAG and IMDB-B datasets, and then used the pretrained compressor on the remaining ones. In Table 3 (left) we report the data as well as the total (data + model) description length, in accordance with the experiments of the main paper. We make the following two observations: (1) As expected, PnC can generalise to similar distributions relatively well (in the

Table 2: Average negative log likelihood of train and test data in bpe (proteins and social network distributions).

| Dataset name | PROTEINS | | IMDB-B | | IMDB-M | |
|---|---|---|---|---|---|---|
| Set | train | test | train | test | train | test |
| PnC + SBM | 3.24 | 3.46 | 0.48 | 0.50 | 0.35 | 0.31 |
| PnC + Louvain | 3.33 | 3.47 | 0.94 | 0.95 | 0.66 | 0.59 |
| PnC + PropClust | 3.41 | 3.53 | 1.43 | 1.61 | 0.95 | 0.75 |
| PnC + Neural Part | 3.33±0.24 | 3.36±0.29 | 1.01±0.05 | 0.99±0.04 | 0.70±0.03 | 0.63±0.02 |

table we highlight the MUTAG $\to$ ZINC and the IMDB-B $\to$ IMDB-M transfer), but fails to do so when there is significant distribution shift. (2) Although MUTAG contains only approximately 100 graphs, it sufficient to train a compressor that can generalise to a significantly larger dataset (ZINC contains approximately 10K graphs), which is an indication that PnC is sample efficient.

Table 3: Out of distribution compression (left) and probability of a subgrpaph to belong in the dictionary (right).

| | Training dataset | | | | | |
|---|---|---|---|---|---|---|
| | MUTAG | | IMDB-B | | same dataset | |
| Test dataset | data | total | data | total | data | total |
| MUTAG | - | - | 6.68 | 7.61 | 2.17±0.02 | 2.45±0.02 |
| PTC | 4.14 | 4.48 | 8.16 | 8.55 | 2.63±0.26 | 2.97±0.14 |
| ZINC | **2.62** | **2.63** | 6.92 | 6.94 | 2.01±0.02 | 2.07±0.03 |
| PROTEINS | 4.74 | 4.87 | 4.31 | 4.44 | 3.34±0.25 | 3.51±0.23 |
| IMDB-B | 1.83 | 1.86 | - | - | 1.00±0.04 | 1.05±0.04 |
| IMDB-M | 1.37 | 1.39 | **0.74** | **0.77** | 0.66±0.05 | 0.72±0.05 |

| dataset | 1 - $\delta$ |
|---|---|
| MUTAG | 0.998 |
| PTC | 0.995 |
| ZINC | 0.999 |
| PROTEINS | 0.999 |
| IMDB-B | 0.995 |
| IMDB-M | 0.997 |

**How frequently do we encounter dictionary subgraphs?** In Table 3 (right), we report the probability $1 - \delta$ for the Neural Partitioning variant of PnC, i.e., the estimated probability of an arbitrary subgraph to belong in the dictionary. Interestingly, since the values are very close to 1, it becomes evident that the partitioning algorithm learns to detect frequent subgraphs in the distribution, which (following Theorem 1b) can in part justify the high compression gains of PnC in all the datasets.

## C.2 Reducing the model size of deep generative models

### C.2.1 Smaller architectures

It is made clear by the experimental results of section 7 that deep neural compression is particularly costly due to heavy overparametrisation. Yet, we also observe that these models achieve strong results in terms of likelihood. Is it possible to strike a better balance between number of parameters and compression cost for a deep generative model? To investigate this, we have conducted the following experiment. We trained 5 GRAN models that differ in parameter count, on 3 different datasets, and monitored the total BPE.

In order to consistently scale the number of parameters across these 5 different models, we have fixed the GNN depth for all models to one, and set for each model the size of the embedding, attention, and hidden dimension, to a constant $c$. Using a different $c$ for each model allows us to explore different scales for the parameter count of the GRAN model. Furthermore, to facilitate comparison with PnC, one of the 5 models is trained without attention and features a reduced amount of mixture components. This is the minimum, in terms of parameter count, working instantiation of GRAN. Finally, we also considered the BPE for the null Erdős-Renyi (ER) model. Figure 1 plots the total BPE of the different models against the number of parameters.

**Results.** In the low parameter regime, the GRAN models are not capable of outperforming the null models and fall significantly behind PnC. At scales that range from $10^3$ to $10^4$ parameters, we observe slight improvements in the total BPE of GRAN on the Proteins and the IMDB-Binary datasets. However, the improved likelihood is not able to compensate sufficiently for the increase in the number of parameters. This becomes more pronounced on larger scales, where GRAN experiences

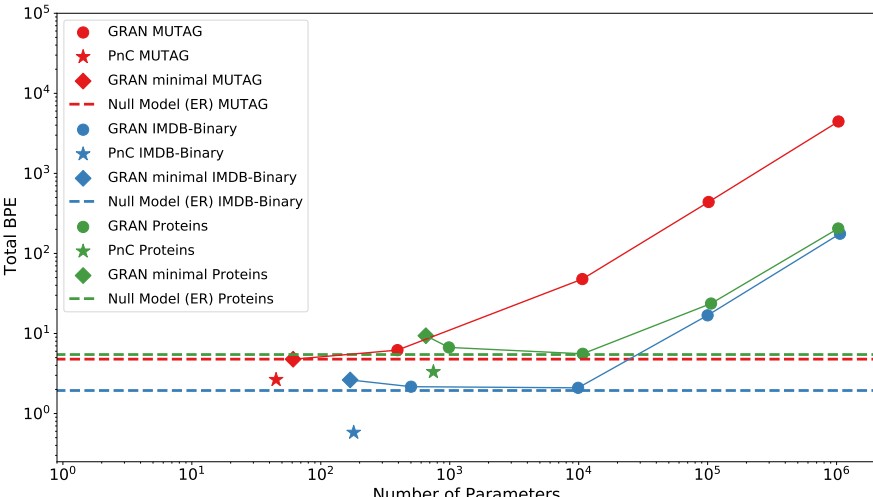

Figure 1: Total BPE of likelihood-based models as a function of the parameter count. *GRAN minimal* refers to the minimum working GRAN model that does not feature attention and multiple mixture components. The non-parametric ER model is represented with dashed lines.

diminishing returns as the cost of parameters outpaces the likelihood gains. On the other hand, the results consistently worsen as the number of parameters grows on MUTAG. In this case, the size of the dataset is an additional detrimental factor that weighs against overparametrised models. Overall, the experiment suggests that, as the number of parameter grows, the off-the-shelf GRAN model becomes increasingly inefficient and is thus not well suited for compression.

### C.2.2 Pruning

Based on the results of the previous section, parameter search alone cannot mitigate the cost of overparametrisation. A more efficient approach to manage the tradeoff between model size and data likelihood is required. However, as shown on table 4, a (at least) two order of magnitude reduction in the model size without a decrease in the data likelihood is required for a more competitive neural compressor with deep generative models. In the following section, we experimented with a combination of modern model compression techniques as a heuristic to reduce the model size.

Model compression techniques aim to reduce the size of a given model while maintaining its performance. Research in model compression has empirically demonstrated that large models can often be considerably shrunk without suffering from major performance losses. Combined with parameter search and mixed precision training, model compression may result in more cost-effective neural compressors. We investigate the feasibility of such an approach in the following experiment.

*First, we decrease the model size by hyper-parameter search.* We fix the depth of both GraphRNN and GRAN to the one provided in the original implementations, and gradually reduce their width to identify a compact version of the network that maintains high performance. This leads to fixing the width of both GraphRNN and GRAN to 16. *Then, we train both models using an iterative weight pruning technique.* We opt for global unstructured L1 weight pruning, using the lottery ticket procedure [19] that has been shown to be effective in the literature. The method we utilised proceeds in the following way: A model is trained for $T$ iterations ($T$ is a hyperparameter chosen based on the convergence and running time of the models on each dataset), then a percentage of the weights are pruned (25% in our case). After pruning, the unpruned weights are reset to their initial state and the process is repeated from the beginning using the new pruned network. This yields up to a 10x reduction in the size of the model in most datasets (we report the best total description length between all pruning phases). *Finally, we attempted to reduce the storage size of the model weights using half precision.* Traditionally, NNs are trained with 32-bit floating point numbers. Recently, progress has

been made in mixed precision training which can enable the use of 16-bit tensors [20]. We follow the same procedure and at the end of training we store the model weights using 16 bits.

Tables 5 and 6 contain the results of both single and half precision pruned models on all datasets. As it can be observed in the results, both models benefit significantly from this hybrid approach, albeit at the cost of reduced data likelihoods. *However, PnC is still able to outperform the pruned versions.* Although our approach to model compression is by no means exhaustive, it becomes evident that the procedure is quite tedious and choosing the right trade-off between the data likelihood and the model size is based on heuristics, hence the minimisation of the total description length cannot be guaranteed. Nevertheless, we believe that this an important research direction that should be further explored in a more principled manner.

Table 4: Minimum model compression ratios required for overpametrised neural compressors to outperform PnC. We assume zero degradation of the data likelihood.

| dataset | GraphRNN | GRAN |
|---|---|---|
| MUTAG | x2980 | x7657 |
| PTC | x995 | x6668 |
| ZINC | x112 | x227 |
| PROTEINS | x105 | x303 |
| IMDBB | infeasible | x1745 |
| IMDBM | infeasible | x2262 |

Table 5: Pruning deep graph generators with single and half precision (molecular distributions)

| dataset name | MUTAG | | | PTC | | | ZINC | | |
|---|---|---|---|---|---|---|---|---|---|
| | data | total | params | data | total | params | data | total | params |
| GraphRNN (half) | 4.70 | 10.77 | 1.08K | 9.53 | 12.10 | 1.10K | 3.89 | 4.10 | 2.64K |
| GRAN (half) | 2.41 | 14.84 | 2.21K | 4.35 | 9.86 | 2.36K | 3.26 | 3.38 | 1.67K |
| GraphRNN (single) | 1.95 | 12.39 | 1.08K | 2.16 | 6.71 | 1.10K | 1.79 | 2.02 | 1.90K |
| GRAN (single) | 2.59 | 24.56 | 2.23K | 4.31 | 14.00 | 2.36K | 3.26 | 3.47 | 1.69K |

Table 6: Pruning deep graph generators with single and half precision (proteins and social network distributions)

| dataset name | PROTEINS | | | IMDB-B | | | IMDB-M | | |
|---|---|---|---|---|---|---|---|---|---|
| | data | total | params | data | total | params | data | total | params |
| GraphRNN (half) | 27.10 | 27.47 | 1.43K | 4.21 | 4.49 | 1.28K | 2.91 | 3.16 | 1.20K |
| GRAN (half) | 3.89 | 4.70 | 3.16K | 0.89 | 1.41 | 2.39K | 0.61 | 1.10 | 2.31K |
| GraphRNN (single) | 2.63 | 3.76 | 2.56K | 1.43 | 1.92 | 1.28K | 0.91 | 1.39 | 1.28k |
| GRAN (single) | 4.28 | 5.11 | 1.78K | 0.84 | 1.75 | 2.38K | 0.55 | 1.41 | 2.31K |

### C.3 Vertex and Edge attributes

Our method can be easily extended to account for the presence of discrete vertex and edge attributes, the distribution of which can also be learned from the data. Assuming a vertex attribute domain $A_V$ and an edge attribute domain $A_E$, we can use the following simple encodings for a graph with $n$ vertices and $m$ edges:

$$\mathrm{L}(\mathbf{X}_V) = n \log |A_V| \text{ and } \mathrm{L}(\mathbf{X}_E) = m \log |A_E|, \tag{15}$$

where $\mathbf{X}_V \in \mathbb{N}^{|V| \times |A_V|}$ the vertex attributes and $\mathbf{X}_E \in \mathbb{N}^{|E| \times |A_E|}$ the edge attributes. One could also choose a more sophisticated encoding by explicitly learning the probability of each attribute.

In this case, the dictionary becomes even more relevant, since when simply partitioning the graph, the attributes will still have to be stored in the same manner for each subgraph and each edge in the cut. Hence, in the absence of the dictionary it will be impossible to compress the attributes.

In Table 7, we showcase a proof of concept in the attributed MUTAG and PTC MR datasets, which are variations of those used for structure-only compression in the main paper. Vertex attributes represent atom types and edge attributes represent the type of the bond between two atoms. As mentioned in the previous paragraph, it is clear that non-dictionary methods are hardly improving w.r.t the null model, which is mainly due to the fact that the attributes constitute the largest portion of the total description length. Another interesting observation is that since the clustering algorithms we used are oblivious to the existence of attributes, they are less likely to partition the graph in such a way that the attributed subgraphs will be repetitive, unless structure is strongly correlated with the attributes. This becomes clear in the PTC MR dataset, where between the different PnC variants, the neural partitioning performs considerably better, since the partitioning is optimised in coordination with the dictionary. In Figure 2 of the main paper, we show the most probable substructures that the Neural Partioning yields for the MUTAG dataset. It is interesting to observe that typical molecular substructures are extracted. This highlights an interesting application of molecular graph compression, i.e., discovering representative patterns of the molecular distribution in question.

Table 7: Experimental evaluation on the attributed MUTAG and PTC MR molecular datasets. **First**, **Second**, **Third**

| Method Family | Dataset name | Atrributed MUTAG | | | Atrributed PTC MR | | |
|---|---|---|---|---|---|---|---|
| | | data | total | params | data | total | params |
| Null | Uniform (raw adjacency) | - | 13.33 | - | - | 16.32 | - |
| | Edge list | - | 12.62 | - | - | 14.06 | - |
| | Erdős-Renyi | - | 9.38 | - | - | 10.87 | - |
| Clustering | SBM-Bayes | - | 9.17 | - | - | 10.61 | - |
| | Louvain | - | 9.37 | - | - | 10.76 | - |
| | PropClust | - | 9.52 | | - | 10.80 | |
| PnC | PnC + SBM | 6.56 | 7.49 | 78 | 8.05 | **9.49** | 158 |
| | PnC + Louvain | 3.52 | **4.45** | 78 | 5.56 | **7.65** | 200 |
| | PnC + PropClust | 5.21 | **6.30** | 54 | 8.51 | 9.58 | 118 |
| | PnC + Neural Part. | 3.83±0.06 | **4.78±0.12** | 74±6 | 5.19±0.39 | **6.49±0.54** | 170±30 |

# D  Implementation Details

**Datasets:** We evaluated our method on a variety of datasets that are well-established in the GNN literature. In specific, we chose the following from the TUDataset collection [21]: the molecular datasets MUTAG [22, 23] (mutagenicity prediction) and PTC-MR [24, 23] (carcinogenicity prediction), the protein dataset PROTEINS [25, 26] (protein function prediction - vertices represent secondary structure elements and edges either neighbourhoods in the aminoacid sequence or proximity in the 3D space) and the social network datasets IMDBBINARY and IMDBMULTI [27] (movie collaboration datasets where each graph is an ego-net for an actor/actress). We also experimented with the ZINC dataset [28–31] (molecular property prediction), which is a larger molecular dataset from the dataset collection introduced in [32]. A random split is chosen for the TUDatasets (90% train, 10% test), since we are not interested in the class labels, while for the ZINC dataset we use the split given by the authors of [32] (we unify the test and the validation split, since we do not use the validation set for hyperparameter tuning/model selection).

**PnC model architecture and hyperparameter tuning:** The GNN used for the Neural partitioning variant of PnC is a traditional Message Passing Neural Network [33], where a general formulation is employed for the message and the update functions (i.e., we use Multi-layer Perceptrons similar to [34]). We optimise the following hyperparameters: batch size in {16, 64, 128}, network width in {16, 64} number of layers in {2, 4}. The learning rate for the updates of the dictionary and probabilistic model parameters was 1 and 0.1 for the fixed partitioning and the neural partitioning variants respectively, while the learning rate of the GNN (neural partitioning only) was set to 0.001. For all the variants we further tune the maximum number of vertices for the dictionary atoms $k$ in {6, 8, 10, 12}. Note that the last hyperparameter mainly affects the optimisation of the Neural Partitioning variant: small values of $k$ will constrain the possible subgraph choices, but will facilitate the network to find good partitions by exploitation. On the other hand, larger values of $k$ will

encourage exploration, but the optimisation landscape becomes significantly more complex, thus in some cases (mainly for social networks, where there is a larger variety of non-isomorphic subgraphs) we observed that the optimisation algorithm could not converge to good solutions.

We optimise each PnC variant for 100 epochs and report the result on the epoch where the description length of the training set is minimum. The best hyperparameter set is also chosen w.r.t the lowest training set description length, and after its selection, we repeat the experiment for 3 different seeds (in total). Table 8 shows the chosen hyperparameters.

Table 8: Chosen hyperparameters for each dataset (PnC + NeuralPart)

| dataset | batch size | width | number of layers | k |
|---------|-----------|-------|------------------|---|
| MUTAG | 16 | 16 | 2 | 10 |
| PTC | 16 | 16 | 2 | 10 |
| ZINC | 128 | 16 | 2 | 10 |
| PROTEINS | 16 | 16 | 4 | 8 |
| IMDB-B | 16 | 16 | 2 | 8 |
| IMDB-M | 64 | 64 | 2 | 8 |

We implement our framework using PyTorch Geometric [35], while the predefined partitioning algorithms were implemented using graph-tool [5] for the SBM fitting and scikit-network [36] for the Louvain and the Propagation Clustering algorithms. To track our experiments we used the wandb platform [37].

**Deep generative models and pruning.** For the generative model baselines, we have used the official implementations provided in the corresponding repositories[3]. For GraphRNN, we trained with the default parameters provided with the official implementation and only tuned the number of training epochs according to the time required for convergence. For GRAN, we adopt one of the configurations provided in the official repository with minor modifications. Namely, we used a DFS ordering, stride and block size 1, 20 Bernoulli mixture components for the parametrisation of the likelihood, and switched of the subgraph sampling feature.

For our iterative pruning protocol, we fix the same number of pruning iterations for both models on each dataset. Specifically we use $\{450, 270, 10, 90, 90, 90\}$ total epochs and a pruning interval $T$ of $\{50, 30, 1, 10, 10, 10\}$ for MUTAG, PTC, ZINC, PROTEINS, IMDB-B, and IMDB-M respectively. We used a 25% pruning percentage, which lead to a 10-fold reduction in model size in most cases. Further pruning was not found to be consistently beneficial in the parameter ranges that we experimented on.

**Model parameter cost.** For the PnC variants we could seamlessly use half precision (16 bits) to store the model parameters (section 4.3 in the main paper) without sacrificing compression quality. However, as discussed in Appendix C.2.2 we were not able to retain similar likelihood estimates when storing with half precision the weights of deep generative models, hence in the results reported in the main tables we used 32 bits to store the model weights[4]. Additionally, regarding the pruned versions of deep generative models, we need to send the locations of the non-zero weights for each parameter matrix in $\mathbb{R}^{d_1 \times d_2}$, which are encoded as follows: $\log(d_1 d_2 + 1) + \log \binom{d_1 d_2}{e}$, where $e$ is the number of the non-zero elements. For all methods compared, the decompression algorithm and the neural network architectures are assumed to be public, hence they do not need to be transmitted.

**Isomorphism.** In order to speed-up isomorphism testing between dictionary atoms and the subgraphs that the partitioning algorithm yields, we make the following design choices: (a) Dictionary atoms are sorted by their frequencies of appearance (these are computed by an exponential moving average that gets updated during training). In this way the expected number of comparisons drops to $O(1)$ from $O(|D|)$. (b) We choose the parameter $k$ to be a small constant value (i.e., does not scale with the number of vertices of the graph), as previously mentioned. It becomes clear, that except for the importance of $k$ in the optimisation procedure, it also plays a crucial role for scalability, since as

---

[3] `https://github.com/JiaxuanYou/graph-generation` & `https://github.com/lrjconan/GRAN`

[4] In a preliminary version of the paper we assumed a 16-bit encoding of the weights without likelihood losses. However, our subsequent implementation and experimentation with half-precision deep graph generators demonstrated that this might not be possible in practice.

mentioned in the introduction in the main paper, solving the isomorphism problem quickly becomes inefficient when the number of vertices increases. (c) Additionally, one can chose to approximate isomorphism with faster algorithms, such as the Weisfeiler-Leman test [38], or with more expressive Graph Neural Networks, such as [11, 10, 39]. These algorithms, will always provide a correct negative answer whenever two graphs are non-isomorphic, but a positive answer does not always guarantee isomorphism. In that case, exact isomorphism can be employed only when the faster alternatives give a positive answer.

## E  Translating probabilities into codes

In the following section, we explain how a partitioned graph can be represented into a bitstream using our probabilistic model. The general principle for modern entropy encoders (Arithmetic Coding [40] and Asymmetric Numeral Systems [41]) is that both the encoder and the decoder need to possess the cumulative distribution function (c.d.f.) of each component they are required to encode/decode. Hence, the encoder initially sends to the decoder the parameters of the model $\phi$ using a fixed precision encoding (e.g., we used 16-bits for our comparisons). The rest of the bitstream is described below:

- **Dictionary.** The dictionary is sent as part of the preamble of the message. It consists of the following:

  (a) the size of the dictionary (we assume fixed precision for this value),

  (b) a sequence of dictionary atoms encoded with the null model, i.e., the message includes the number of vertices, the number of edges and finally the adjacency matrix: $k_i, m_i, E_i$ (see Eq. (2) in the main paper).

- **Graphs:**  Subsequently, each graph is sequentially transmitted. The message contains the following:

  (a) the total number of subgraphs and the number of dictionary subgraphs $b$ and $b_{\text{dict}}$ that are encoded using the parameters of the categorical distribution $q_\phi(b)$ and the binomial distribution $\text{Binomial}(b_{\text{dict}}|b; \phi)$. The c.d.f. of the binomial distribution can be computed using a factorisation described in [42],

  (b) the subgraphs that belong in the dictionary, which are encoded using the multinomial distribution $q_\phi(\mathcal{H}_{\text{dict}}|D)$. As above, a factorisation described in [42] can be used to compute the c.d.f.,

  (c) the non-dictionary subgraphs. These are encoded with the null model (same with the encoding of dictionary atoms as mentioned above),

  (d) the cuts, which are encoded using Eq. (8).

Several of our encodings involve uniform distributions over combinations of elements (e.g., for the adjacency matrix in the null model). To compute them, we can either factorise the distribution as in [43] in order to efficiently compute the c.d.f, or use a ranking function (and its inverse for the decoder) that maps a combination to its index in lexicographic order (e.g., see [44]).