# OpenReview forum: "Partition and Code: learning how to compress graphs"
_NeurIPS.cc/2021/Conference — NeurIPS 2021 Poster_

### Official Review · Reviewer_Q2XE · 2021-07-04

**Rating:** 5
**Confidence:** 4

**Summary:**

The paper studies the lossless compression of graphs with  deep learning. The authors propose to compress  the graphs in 3 steps: a partitioning algorithm decomposes the graph into elementary structures, which are mapped to the elements of a small dictionary, and an encoder translates the representation into bits. The partitioning  is parameterized  by a graph neural network output the likelihood of partitioning and optimized with the policy gradient method to achieve as small minimum description length as possible. The minimum description length are decomposed into several parts, the dictionary and encoding etc. A few experiments are conducted on small graph datasets to show 30% space compression  compared to the naive edge list, and 50% compared to the Erdos-Renyi method.

**Main Review:**

The paper studies the lossless compression of graphs with  deep learning. The authors propose to compress  the graphs in 3 steps: a partitioning algorithm decomposes the graph into elementary structures, which are mapped to the elements of a small dictionary, and an encoder translates the representation into bits. The partitioning  is parameterized  by a graph neural network output the likelihood of partitioning and optimized with the policy gradient method to achieve as small minimum description length as possible. The minimum description length are decomposed into several parts, the dictionary and encoding etc. A few experiments are conducted on small graph datasets to show 30% space compression  compared to the naive edge list, and 50% compared to the Erdos-Renyi method.

Overall, this paper presents a well-executed though not surprising method for compressing graphs by learning and exploiting the most common subgraph structures of the dataset. We can save space by using parameterized graph neural network and policy gradient to optimize the encoding length. The proposed method has many steps and obviously it takes time and effort to make sure the engineering eventually works out. It's interesting and in fact impressive to see authors can make such complicated engineering work out. On the down side, there are a few concerns listed below. In any case, my general evaluation is that this paper though not surprising or exciting, still has some values, and hence is borderline for acceptance, that is, it may be accepted or rejected depending on the meta context.

Some detailed concerns:

0. The datasets compressed are quite small so I'm not sure  why authors want to compress them.  Can you show  compression performance on datasets with > 300 GB? In my opinion, this is the most important technical weakness of the paper.

1. The paper is generally easy to read, but it leaves  pretty  much all  details of the algorithm in the appendix. This makes it almost not possible to understand most details of the algorithm. Some questions: a) how exactly the graph partitioning  is parameterized by graph neural network, b) how the cuts connecting subgraphs are represented, c) how to construct dictionary and assign encoding, d) how to exactly connect the subgraphs and write  that in the bits, e) how graph isomorphism is conducted to  further save space.  A pseudo code should also be provided in main text.  The main ideas of these should all be described in the main text, possibly with pictures to better illustrate the algorithm steps.

2. The proposed method is quite complicated engineering, so authors should prepare more details describing how to make such complicated engineering actually work out. This is important for reproducibility as it's imaginable that readers will not be able to reproduce without more practical guidance, which will heavily limit the usage and influence of this paper, as most researchers obviously will not be interested in reading and applying complicated engineering papers that are not easy to follow and reproduce.

4. I don't see any theoretical insights and depth in this paper. It's fine for practical papers, but I feel  authors can do better in this regard. What are the potential limitations and guarantees? Can you prove a lower and upper bound? Can you show tightness of the bound?

5. Missing related work that you may want to cite and discuss:

Solving graph compression via optimal transport. NeurIPS 2019.

Learning-Based Low-Rank Approximations. NeurIPS 2019.

Learning Sublinear-Time Indexing for Nearest Neighbor Search. ICLR 2020.

**Time Spent Reviewing:**

5

---

> ### Author Response · Authors · 2021-08-10
> **Official author reply to Reviewer Q2XE**
>
> We thank the reviewer for their evaluation and their suggestions to improve our manuscript. However, we think there is a misunderstanding regarding our main contributions and the technical parts/implementation details of our work. In order to rectify this, below we emphasise the fundamental challenges of the problem (which in our opinion are far from trivial) and how our framework addresses them for the first time.
>
> ### 1. Our contributions
> As explained in the main text, the absence of ordering (or equivalently graph isomorphism) makes the problem of graph compression highly non-trivial and significantly different from what is known for conventional ordered data (images, text, etc.). At the same time, to the best of our knowledge, the problem remained completely unexplored from the machine learning community. Although some efforts have been done from other communities, there was no information-theoretic based literature aiming at compressing arbitrary graph distributions (the only methods available focus on simple distributions such as the Erdos-Renyi model), while the majority is based on engineered codecs lacking any guarantees on the expected description lengths.
>
> On the contrary, our work is the *first* to:
>
> 1.  put forward the necessary principles and challenges of this fundamental problem,
> 2. provide a framework for data-driven graph compression that is applicable to arbitrary graph distributions and its compression quality is theoretically guaranteed,
> 3. provide a practical methodology to minimise the description length objective, including solutions to a variety of also relatively unexplored sub-problems, such as:
>     - learning to partition,
>     - graph dictionary learning,
>     - neural compression without overparametrised generative models.
>
>
> ### 2. Dataset sizes
>
> First, let us discuss two different types of datasets w.r.t. scalability:
>
> 1. Larger datasets with small or medium-sized graphs (e.g., large molecular databases). This type would not be an issue for our method. More training time will be required, and our method would benefit from further optimisations in the implementation, but the compression ratio is not expected to change much from what we report in the Tables (e.g., observe that MUTAG and ZINC have similar compression rates, although the former is two orders of magnitude smaller than the latter),
>
> 2. Datasets with large/huge graphs. Here scalability might be an issue, as also mentioned in the paper. We do acknowledge in the paper that this is a limitation, but we expect it to be addressed in the future with e.g., parallelisations (as in RNNs for NLP when processing large texts), i.e., sampling multiple subgraphs instead of one at each iteration, or other partitioning algorithms. Note that is mainly an issue when the partitioning algorithm is trainable since the partitioning needs to be repeated at each epoch. However, as Tables 1 and 2 show, in some cases, a non-parametric partitioning that needs to be run only once (e.g., PnC + SBM), performs remarkably well, which mitigates the scalability issue.
>
> It is true that the datasets we considered are not as large as the size of 300GB that the reviewer mentioned. However, given the fundamental challenges of the graph compression problem that were currently completely unaddressed, in our opinion, this should not be considered as a big weakness at this stage.
>
> ### 3. Algorithmic details in the appendix \& implementation details
>
> The reason why these algorithmic details were deferred to the appendix, besides the page limit, is that we wanted to emphasise the high-level rationale of the framework without making the main paper heavy for a reader that is not familiar with the topic. However, all the questions the reviewer mentioned are analysed in detail in the supplementary material. As mentioned in our reply to all reviewers, we will provide a more detailed explanation of the partitioning algorithm in the main text.
>
> Regarding reproducibility, Appendix D provides all the implementation details needed for the interested reader to reproduce our results. Moreover, of course, our code will be made publicly available. We understand that the reviewer might be sceptical due to the fact that we are optimising over 3 sets of parameters and the problems involved are of combinatorial nature. However, no detail is obfuscated and we made sure to explain our algorithm both from an intuitive viewpoint and at a low level.
>
> ### 4. Theoretical insights
>
> As mentioned in our response to all reviewers, this is an excellent point raised that we were also curious about. To that end, we performed an in-depth theoretical analysis providing deeper insights on how different compressors are expected to perform and showing clear quadratic and linear gains of PnC against other baselines. Through this analysis, it became evident why both partitioning and dictionary learning are nearly always a good idea.
>
> ### 5. Related work
>
> Regarding the references you mentioned:
>
> -  Ref [1] is related to lossy graph compression and graph summarisation, rather than lossless compression (as we focus on here). The authors show how to convert the graph to a smaller, compressed one. However, the original graph cannot be reconstructed and it is not clear how to obtain a binary representation.
>
> -  Ref [2] is also lossy and it is not clear if it can be applied to adjacency matrices (with 0,1 entries), while the problem of isomorphism (challenge C1) cannot be addressed. Hence, we do not see any obvious connection with our work.
>
> - In [3] the authors leave an open question regarding optimising graph partitioning jointly with learning a classifier, which is a problem that we do address in our work. Apart from this, we could not locate an immediate connection with our work.
>
> If we have missed a connection that the reviewer would like to suggest, we would be of course eager to highlight it in our manuscript.
>
> [1] Solving graph compression via optimal transport. NeurIPS 2019.
>
> [2] Learning-Based Low-Rank Approximations. NeurIPS 2019.
>
> [3] Learning Sublinear-Time Indexing for Nearest Neighbor Search. ICLR 2020.

---

> > ### Comment · Reviewer_Q2XE · 2021-08-11
> > **Final review update**
> >
> > After reading the author response, I decided to keep my original score, weak reject.
> >
> > While the addition of some theoretical insights is good, my other concerns were not addressed.
> >
> > Only small datasets were evaluated so this really limits the usefulness of the paper. While the authors tried to find some excuses for not using large datasets, this makes the paper not so attractive. In fact even 300GB as I suggested is not very large. In practical applications, we usually have to deal with 20 TB datasets. So  I don't really see much value in compressing small graphs.
> >
> > While authors tried to emphasize  the contributions again, I would like to point out that I understand these contributions, but I just find the paper not exciting. It's also quite bizzare that in the response authors claim the mentioned related works are not related ...

---

> > > ### Author Response · Authors · 2021-08-14
> > > **Concerns regarding justification of rating**
> > >
> > > We respect the reviewer's opinion, but we are concerned that the justification of the rating is not sufficiently backed by scientific arguments.  As a final remark we would like to stress the following:
> > >
> > >  - Excitement is inherently subjective and thus presents weak scientific grounds for rejection. Indeed, the opinion of the other reviewers about the importance of our contributions is directly contradicting this reviewer's opinion.
> > >
> > > - Further, we respectfully disagree with the reviewer’s argument that the lack of evaluation on datasets exceeding 100s of GB is grounds for rejection, for a number of reasons:
> > >    - There is no fundamental obstacle in compressing datasets with more graphs with PnC. Our compressor can be applied to an arbitrary number of test graphs (additionally we can train a compressor with relatively few samples - see our response to reviewer dgax) and unless there is a significant distribution shift, the compression rates will be the same.
> > >   - Note that many labs do not have access to the amount of resources usually required to conduct such large-scale studies (20TB datasets), especially in the small time window available for rebuttal. Hence penalising the problems emerging from this inequality should be done cautiously.
> > >   -   Regarding compressing larger graphs, our theoretical results imply that our method would benefit even more for graphs with larger size $n$, hence the only current obstacle is designing more efficient implementation techniques to improve scalability.
> > >   -  In our paper, as well as in our response above, we acknowledge this limitation and discuss potential ideas to improve scalability.  We strongly believe that scientific progress, including many papers accepted in ML conferences, does not necessarily require solving every single aspect of a problem, especially when the problem has not been explored in the past (as in our case). Hence penalising the existence of a limitation that we do not claim as a contribution of our work is, in our opinion, unfair.
> > >   -  Moreover,  the reviewer mentioned that “in practical applications, we usually have to deal with 20TB datasets”,  but no concrete suggestions were made, which makes this comment less actionable (any particular pointers are welcome).
> > >
> > > - Finally, regarding the references pointed by the reviewer, if the reviewer does not agree with the connections we identified (or rather the lack of), we would expect a more concrete explanation.

---

> > > > ### Comment · Reviewer_Q2XE · 2021-08-14
> > > > **Response to the author's claim "Concerns regarding unscientific grounds for rejection"**
> > > >
> > > > While the final decision will be left to the other reviewers and area chair, here I briefly respond to the authors claim  "Concerns regarding unscientific grounds for rejection".
> > > >
> > > > As in the review, for practical papers, the "excitement" comes from demonstrating practical usage of the method, i.e., successfully compressing large datasets with large graphs, and hopefully uncompressing them for downstream tasks. The authors did not demonstrate such practical value so there is no excitement. Specifically, since the datasets in the paper are quite small, practitioners will likely not be convinced to spend time studying and applying the complicated method for their applications. Is this clear? Of course practical value is not the only aspect one could get excited  about a paper, other examples are interesting theoretical insights and proof techniques. Authors could demonstrate any of these aspects to make the paper exciting. Still, as I mentioned in the review, this paper has some value but is borderline. There has never been any "grounds for rejection". Rather, we seek exciting aspects of a paper that makes one eager to study, apply, and accept the paper.  Hope this helps.

---

### Official Review · Reviewer_u2r2 · 2021-07-06

**Rating:** 8
**Confidence:** 3

**Summary:**

This article presents a procedure to compress graphs (into a code). It takes the principle of entropy minimization i.e. finding a code that has the lowest possible entropy. In order to do so three steps are considered: 1) partioning the graph into subgraphs ($\operatorname{Part}_{\theta}$ which is parametrized by a NN) 2) finding a dictionary that faithfully describes a prior on the subgraphs structures, and that will be used to encode the different subgraphs coming from PART 3) Graph encoding part: assign a (parametrized) probability to each subgraph (which depends of the dictionary also). From a probability we can derive the length of the code. Overall this defines a description length which can be minimized in order to have the shortest code possible.


**Limitations And Societal Impact:**

Authors did not provide any potential negative societal impact of their work however, it is difficult for me to foresee these impacts... Some other limitations are addressed in the supplementary.

**Main Review:**

First a disclaimer: I am not an expert on information theory, especially in the context of graphs, so I may have missed some details. In the following I will note from (1/5) for a comment that have a small impact on my final score to (5/5) for a comment that has a strong impact.

- Pros:
	- Overall the article is very well written and enjoyable to follow.
	- The problem tackled is quite interesting and far from trivial. I think the article manages to give good insights and paves the path for new perspectives.

- Cons:
	- Some parts of the article are a bit vague/unclear in my opinion and would deserve more details.
	- There are a lot of "black-boxes" so that it is sometimes hard to understand the intuitions behind the different mechanisms.
	- I am not really convinced by the usefulness of the "dictionary part".

Overall I find that the article is very well written, at least up to Section 5, and enjoyable to follow. I thank the authors for that. The different problematic are clearly stated as well as the different "blocks" of reasoning.  I like the idea of relating information theory with ML in the precise context of graphs. The "commun graph encodings" scheme seems reasonable to me, since, as far as I known, the Erdos-Renyi assumption (used to define $L_{\operatorname{null}}$) is one of the few generative process of graphs where we can say something about optimal compression (ref [9]). The fact that the estimated distribution must be invariant to isomorphism is sounded (row 223). For this argument I think Appendix A is quite enlightening and really shows that one needs to encode the isomorphism classes. It is a bit of a shame that this discussion is not included somehow in the main text or at least more detailed (0.5/5).

- About the universe and the dictionnary:

(4/5) It is not very clear to me how the approximation made by considering eq (6) is "sufficient" for solving the challenge C.1, ie only considering isomorphism with respect to atoms in the dictionary. I can be wrong but it seems that the case $\exists a_i \in D$ such that $H_i \sim a_i$ ($\sim=$ isomorph) is hardly met in practice when the size of $D$ is reasonable. Indeed, for me it is unlikely to find a graph within the dictionary that is isomorphic to a subgraph coming from a black-box GCN used in PART. Or at least it is not clear how often this occurs so that I expect $q_{\phi}(H_i|D)=q_{\operatorname{null}}(H_i)$ everytime; especially in the case of "small" universe. Also it is written in row 284-285 that "most subgraphs (in the universe) will never be encountered in the graph distribution": this sentence is a bit confusing, because it seems to corroborate my remark. Overall I am quite skeptical regarding the usefulness of the dictionary and thus the ability of the method to deal with isomorphisms.

(4/5) More importantly how, in practice, do you choose this "practically enumarable universe" (row 277) ? I do not see any details about this in the article.

- About the encoding:

(2/5) It is not so clear to me how $q_{\phi}(a_i)$ are defined. It is said that (row 276) "are parametrized by learnable variables that are converted into categorical distributions [...] using a sofmax function". What do you mean by "learnable variables" ? Does it boil down to an optimization on the probability simplex (whose size is the number of elements in the dictionary) ? Or do you choose some parametric family like binomial and learn the different parameters this way ?

- About the partitioning algorithm:

(3/5) I think a more detailed discussion about the partitioning algorithm is welcomed here. In this form, $\operatorname{Part}_{\theta}$ is a kind of "magic" black-box and we have no idea how the REINFORCE algorithm works. Even if some detailed are given in the supplementary I think it is important to give at least some intuitions behind the PART algorithm and other potential non-parametric strategies (to understand for example the complexity of it). Is it really stable ? Can two runs of the same algorithm give a very different code ?

- About the decoding:

(3/5) I do not understand also how one could potentially decode from a compressed graph here. The authors present well the stakes of the information theory with the aim of creating a uniquely decodable code (row 148: " the code needs to be uniquely decodable"). However with the many models and approximations on the true distribution (which seem to be essential because the different problems are NP-hard) I think the resulting code has no reason to be uniquely decodable. How in practice we can recover the graph from the code ?

(1/5) Related to this remark I think it would have been valuable to detail more what is "jointly exchangeable" here, even with one or two sentences.


- Experiments:

(4/5) I find a bit of a shame that no comparison are made with algorithms that provide compression codes with guarantees. How does it compare for example with a "simpler" algorithm such as Structural zip (SZIP) (ref [9] of the paper) which assumes that the underlying graph distribution is Erdos-Renyi or some other ones described in ref [32] of the paper ? Also I think a small ablation study regarding the usefulness of the dictionary here could be interesting (see my point above about $q_{\phi}(H_i|D) \approx q_{\operatorname{null}}(H_i)$ everytime).


- Other comments:
	- I think in Section 4.5 authors talk about eq (8) and not eq (7)


----- AFTER REBUTTAL -----

The authors fully addressed my concerns I think the paper is very rich and deserves publication (see my comment below). I change my score to 8.

**Time Spent Reviewing:**

5

---

> ### Author Response · Authors · 2021-08-10
> **Official author reply to Reviewer u2r2 (1/2)**
>
> ### 1. The importance of the dictionary
>
> * *Why does the dictionary make sense? Because the subgraphs are small:* First a clarification: Completely dealing with challenge C1., i.e., encoding the isomorphism class of a graph, would require (1) solving graph isomorphism (which is NP-intermediate) and (2) a large number of training samples. Hence our solution, i.e., using a dictionary, is a "middle-ground" between optimal compression and tractability (both computational and statistical). This is more formally stated in Theorem 1 in the appendix. Observe that the bits saved are $\Theta(n \log k)$, where $k$ is the maximum size of the graphs in the dictionary, while completely dealing with C1 requires saving  $\Theta(n \log n)$ bits. This theoretical evidence will be added in the main text, together with the results reported in the reply to all reviewers.
>
> The issue raised by Reviewer u2r2 regarding the importance of the dictionary would be indeed valid if the subgraphs were "large", i.e., proportional to the size of the graph $O(n)$. In that case, the statistical argument mentioned above can be invoked, which implies that the subgraphs in the dictionary would be rarely encountered in a test graph, unless the training set is too large.
>
>  On the contrary, this is not an issue for our method and the justification lies exactly in our solution to decompose the graph in small subgraphs of size $k=O(1)$. These are usually elementary subgraphs that are common building blocks of graph distributions and are very frequently encountered within real-world graphs. For example, observe the subgraphs in Fig.2: as can be seen the dictionary may contain simple graphs such as paths, cliques and cycles.
>
> * *Learnable dictionary:* Reviewer u2r2 mentioned: "for me, it is unlikely to find a graph within the dictionary that is isomorphic to a subgraph coming from a black-box GCN used in PART." Recall that that dictionary is learnable, hence its elements will be decided according to the choices of the partitioning algorithm and not before observing the data. Let us also clarify the phrase "most subgraphs in the universe will never be encountered in the graph distribution": We define the universe as "all graphs of size up to $k$". However, instead of pre-defining this set, we build the universe adaptively by only consider subgraphs (with size up to $k$) that the partitioning algorithm yields. This is an implementation trick that allows us to do fewer graph isomorphism comparisons and is not related to the discussion above.
>
> - *Role of GNN:* Finally a side note. Since GNNs have inherent limitations in fully exploiting the graph structure, the partitioning will be usually biased towards subgraphs that are easier for the GNN to detect. Although this might lead to sub-optimal solutions, we argue that the GNN-based partitioning acts as a surprisingly well-performing heuristic, selecting subgraphs that lead to small description lengths.
>
> - *Empirical evidence:* To be more convincing, below we provide the values of the probability that a subgraph belongs in the dictionary ($1-\delta$) for all the datasets. Observe that this value is usually close to 1.
>
> |dataset| 1-$\delta$|
> |---|---|
> |MUTAG| 0.9969|
> |PTC| 0.9909|
> |ZINC| 0.9994|
> |PROTEINS| 0.9999|
> |IMDBBINARY|0.9871|
> |IMDBMULTI| 0.9967|
>
>
> Finally, the results in Tables 1 and 2 between the pure partitioning based methods and PnC showcase the importance of the dictionary: in every case PnC provides better compression of the data.
>
>
> ### 2. Enumerability of the universe
>
> As mentioned above, the universe is built on the fly, hence in principle enumerability is not an issue either. Implementation-wise, we predefine an arbitrarily large maximum universe size $|\mathfrak{U}|$ in order to initialise the membership variables $\hat{x}_i$.
>
>
> ### 3. Atom encoding
>
> We thank the reviewer for pointing out this vague point. Indeed it boils down to an optimisation on the probability simplex of the dictionary. This is relaxed in Eq. (2) of the supplementary, in order to account and optimise for the membership variables of the dictionary as well.
>
> ### 4. Partitioning
>
> As mentioned in our reply to all reviewers, some details of the partitioning algorithm will be brought forward from the supplementary to the main text. Regarding other non-parametric strategies, any graph decomposition algorithm can be used (such as the ones in Table 1 and 2, e.g., SBM fitting, Louvain algorithm, clustering by label propagation). Additional alternatives are discussed in the supplementary. The complexity of our proposed partitioning algorithm is $O(n)$, since vertices are sampled iteratively. We will clarify this in the main text.
>
> Regarding the stability of the algorithm, in the supplementary material, appendix C.3., table 2, we repeated our experiments with multiple seeds showing that even in the presence of extra stochasticity the variance is relatively low. During the rebuttal phase we also tested the stability only w.r.t. different sampled partitions for all the datasets and observed that the variance is close to 0-0.1 bpe. This is probably due to the fact that the REINFORCE algorithm becomes overconfident after a certain number of iterations. To conclude, the algorithm *can* potentially yield a different code for the same graph (although with small probability - we assume that this will be more common for graphs with a large number of non-trivial symmetries), but this is not an issue as we discuss next.

---

> > ### Author Response · Authors · 2021-08-10
> > **Official author reply to Reviewer u2r2 (2/2)**
> >
> > ### 5. Unique decodability
> >
> > Let us elaborate on the unique decodability property, hoping to clear up confusion:
> >
> > 1.  *Code redundancy vs unique decodability*: A code is redundant when two different codewords are mapped to the same symbol (the same graph in our case). For example, if the partitioning algorithm yields two different decompositions of the same graph, then we will end up with two different codewords that describe the same object. Unique decodability means that a codeword can be translated to a single symbol, i.e. there is no ambiguity in the decoding. Hence, redundancy does not compromise unique decodability, it just implies that our code is not optimal (which is expected, otherwise we would be able to solve isomorphism with our compression algorithm).
> >
> > 2. *How to ensure unique decodability?* Every code that is derived from a probability distribution is uniquely decodable (this is an informal explanation of the Kraft-McMillan inequality of Eq.(1)). This is precisely what entropy coders do: they receive a probability distribution and then translate a probability into code with approximate codelength of $-\log p$. Both the encoder and the decoder possess the parameters of the distribution and hence can both map codewords to symbols.  In our case, each component of the graph encoding is derived from a probability distribution, hence it is uniquely decodable, and each codeword can be decoded one-by-one by processing the bitstream from left to right.
> >
> > A detailed explanation of the encoding-decoding of the different components can be found in the supplementary material, Appendix E. After decoding each component, we end up with the collection of subgraphs $\mathcal{H}$ and the cuts that connect them $C$. To see why this is sufficient to completely reconstruct the graph, let us use an explanation via the adjacency matrix $A$ of the graph. Consider an arbitrary ordering of the elements of $\mathcal{H}$. Then, each element $H_i$ corresponds to a diagonal block of the adjacency matrix $A_{ii}$. Subsequently, we use the same ordering to order the elements of $C$. Then each element $C_{ij}$ corresponds to an off-diagonal block $A_{ij}$. Hence, we have reconstructed an adjacency matrix of the graph which is a complete representation.
> >
> > ### 6. Jointly exchangeable property
> >
> > We thank the reviewer for their suggestion. We will add a more formal explanation in the updated version. Briefly, by jointly exchangeable we mean that the probability $q_{\phi}(\mathcal{H}, C)$ should be the same for every permutation $\pi$ that acts jointly on $\mathcal{H}$ and $C$, i.e., $q_{\phi}(\mathcal{H}, C)  = q_{\phi}(\pi \cdot \mathcal{H}, \pi \cdot C)$, equivalently:
> >
> > \begin{equation}
> > q_\{\phi\}\big((H_i)_\{i\in[b]\}, (C_\{i,j\})_\{i,j\in[b]\}\big) =  q_\{\phi\}\big((H_\{\pi(i)\})_\{i\in[b]\}, (C_\{\pi(i), \pi(j)\})_\{i,j\in[b]\}\big),
> > \end{equation}
> >
> > where $[b] = \{1,2, \ldots b\}$.
> >
> > ### 7. Comparison with SZIP or other algorithms
> >
> >
> > We agree that it would be beneficial to compare with other graph compression algorithms that have been developed outside the machine learning domain. Indeed the SZIP algorithm might be a strong baseline, since the authors claim to be able to approach the entropy lower bound for *unlabelled* Erdos-Renyi graphs which in principle could save $\Theta(n \log n)$ bits. However, in our attempt to compare with this method we encountered a number of obstacles: (1) to the extend of our understanding, the theoretical description length provided by SZIP is not explicitly computable (simply subtracting $n \log n$ bits from the labelled Erdos-Renyi baseline will be too optimistic, since  additive superlinear terms will be ignored - see Theorem 2 of the SZIP paper) , hence in order to obtain the description lengths we need an exact implementation of their algorithm paired with an arithmetic encoder. However, (2) there is no publicly available code and the algorithm is non-trivial to implement.
> >
> > Regarding other algorithms mentioned in [32], the most relevant ones are those based on vertex reordering. Similarly here, we could not find any method with explicitly computable theoretical description lengths, while most of these methods either do not provide publicly available code, or the code is not maintained/is hard to re-implement from scratch. However, the partioning-based baselines are similar in spirit with vertex-reordering and allow for explicitly computable description lengths, hence we believe that they are good representatives of this family of methods. Of course, we are open to suggestions from the reviewers and in case a method that doesn't suffer from the above two issues exists, we would be happy to compare for completeness.

---

> > > ### Comment · Reviewer_u2r2 · 2021-08-17
> > > **Response to authors**
> > >
> > > I would like to thank the authors for their very detailed response that was very helpful for me to understand the unclear points. More precisely my concerns about "About the universe and the dictionary" and "About the decoding" were fully addressed. It is still a shame that the Partition algorithm and Jointly exchangeable property are only detailed in the supplementary, but this is due to the space limitation (and maybe details can be added given the extra-page if the paper is accepted). I also acknowledge that SZIP is not a direct competitor and that it is difficult to apply in this setting. Giving that many other comparisons are made in this paper I do not think it is a major problem at all.
> > >
> > > Overall, after having read the different comments and after having understood the paper more deeply I think this work is very rich and paves the path for many interesting applications and theoretical works. I am changing my score accordingly.

---

### Official Review · Reviewer_dgax · 2021-07-13

**Rating:** 7
**Confidence:** 2

**Summary:**

The authors propose a lossless compression method for graphs. By analogy to patches in images, or words in text, they propose breaking up the graph into sub-graphs, and encoding common sub-graphs using a dictionary. This is then compressed by an entropy encoder. Experiments are provided on a variety of datasets, ranging from molecules to social networks

**Limitations And Societal Impact:**

I don't think the limitations are sign-posted well enough. It'd be useful to provide some discussion about this somewhere. I don't think the authors have done a good enough job of ablating their work, since there are many interesting questions that can be asked.

**Main Review:**

*Disclaimer*: While I am familiar with information theory and compression, I certainly would not rank myself as an expert. My expertise is more to do with graphs.

### Detailed Comments

Abstract: This is good, although it'd be even better with some numbers.

Intro: This is also well written, and explains the fundamental idea quite well, breaking it up into the constituent parts.

Related Work:
- L90: it's worth noting that vertex reordering can be very slow to do, so even if it were to provide solid compression, it's not going to be fast.
- L94: Surely the premise here is that we are learning a (limited) graph grammar, so in some sense this is related?
- L124: Typo for 'therefore'

Preliminaries:
- L143: It is interesting that this can compress families of graphs. Arguably, however, this means the size of the compression model is less important though? If you can generalise to unseen graphs, then that's also useful.
- I thought this section was fairly well written; you may wish to expand the explanation of eq. 1 though.

S4:
- I think you could certainly do with having a diagram explaining the overall process around here. The prose is extremely formal, and obscures much of the intuition.
- L247: you should forward reference the next section here, by saying that you'll explain how to optimize it next. This seems rather abrupt and odd as currently written.

S5:
- L295: I am not sure I agree that leaving the partitioning algorithm to the appendix is a good idea if you're selling it as part of the novelty of the paper. The prose is overly verbose and formal: I am certainly you can compress some equations and re-write some sections so you have enough space that this does not need to be relegated.

S6:
- Fig2 is interesting in that individual nodes are considered a probable subgraph for both datasets (although, this is also not surprising); how common are they in practice? Do subgraphs commonly get broken up into this most primitive atom?
- Table 1/2: the results are good, but there's no indication of training time here (for any method), which should not be ignored.
- If you were going to provide another dataset, it might be good to look at something like OGB code? (note: I don't expect you to do this. But just a thought for future :-) )

#### Other Questions:
- An interesting property of this work is that it should generalise to graphs which are "unseen" (just as an image compressor can generalise to images that haven't been seen before). Could you evaluate this property? Maybe you could split the datasets, and report the compression rate on the holdout data?
- Similarly, what happens if you take a compressor and apply it to a dataset it wasn't trained on? How well do things transfer?
- How much data do you need to train a reasonable compressor? Would it be possible to plot a graph?
- Finally, how do you handle node / edge features?

I'm not expecting all of these to be answered necessarily in the time you have for rebuttal. But it'd definitely be useful to answer some of them :-)

#### Overall Thoughts

I think the work is really interesting, but the evaluation doesn't do it complete justice. For now it's a weak reject, but with some work done at rebuttal I am happy to increase the score.

===== Post rebuttal

I'm happy to increase my score (although I stay unconfident in my assessment). Another reviewer did make a good comment about scaling to larger datasets. I think this point is true. Given that the authors are able to show that your method generalizes to the test set/across datasets, I think this is not too difficult for the authors to address given some time.

**Time Spent Reviewing:**

3

---

> ### Author Response · Authors · 2021-08-10
> **Official author reply to Reviewer dgax**
>
> We are thankful to the reviewer for their suggestions to improve our manuscript. All typos will be corrected and minor suggestions regarding the exposition of the ideas will be incorporated in an updated version. Below we address your comments one by one:
>
> ### 1. Partitioning algorithm in the main text
>
> As mentioned in our response to all reviewers, we will do our best to fit it into the main paper.
>
> ### 2. Individual nodes
>
> This is a very interesting observation. In practice, we observed individual nodes to be (almost) always included in the dictionary, and are mapped to high probability values (from ~5\% to ~25\%). There are two reasons why this happens:
>
> * Hub-like vertices: The cut between a hub and other subgraphs in the partition is “low-surprise”, i.e., either the hub is connected to all the vertices of another subgraph or it is completely disconnected. In that case, the hub improves the compression rate and it is selected by the neural partitioning algorithm.
>
> * Isolated vertices might show up after the removal of larger subgraphs from the graph. For example, our neural partitioning algorithm iteratively decomposes the graph into subgraphs by first predicting the number of vertices in each subgraph and then the vertices themselves. Hence, a slightly suboptimal prediction on the subgraph size will result in isolated vertices. Although less pronounced, a similar phenomenon might be observed for individual edges.
>
> ### 3. Node and edge features
>
> PnC can compress graphs with discrete vertex and edge attributes. In the supplementary material, section C.1, we provide a detailed description of the methodology. Briefly, every vertex and every edge of a subgraph (either dictionary or non-dictionary) is employed with an extra codeword representing its vertex or edge attribute respectively (in the simplest case this can be of length $\log|A_V|$ and $\log|A_E|$, where $A_V$ and $A_E$ the number of unique vertex and edge attributes). The same procedure is done for the cuts with regards to their edge attributes.
>
>
> ### 4. Ablation studies
>
> **Training time.**  Indicatively, for the PnC + Neural Part. compressor we have the following training/inference times per graph (batch size 16, $\bar{n}$: average number of vertices):
>
> MUTAG ($\bar{n} = 18 $, training graphs: 169): ~0.035/0.025 sec,
>
> PTC ($\bar{n} = 26$, training graphs: 309): 0.056/ 0.034 sec,
>
> PROTEINS ($\bar{n} = 39$, training graphs: 1001): 0.104/0.073 sec.
>
> These lead to around ~10min, ~29min and ~3h total training times for 100 epochs. Note that the neural partitioning iteratively selects vertices from the graph, so it scales with $O(n)$. A detailed table, including training and inference times for all trainable methods may be included in the supplementary material.
>
> **Generalisation to unseen data.** Indeed, our graph compressor can generalise to unseen data and this is an important property that should be highlighted in the evaluation. Since the generalisation gap was usually small, in Tables 1 and 2 we show the compression quality on the entire dataset (training + test set). Indicatively for the PnC + Neural Partitioning model, the corresponding train and test compression rates (without taking into account the model size) for the experiments shown in the tables are the following:
>
> |dataset|train (bpe)| test(bpe)|
> |---|---|---|
> |MUTAG| 2.36| 2.49|
> |PTC| 2.72| 2.77|
> |ZINC| 2.19| 2.20|
> |PROTEINS| 3.14| 3.20|
> |IMDBBINARY| 1.06| 1.03|
> |IMDBMULTI| 0.73| 0.84|
>
>
> Similar trends are observed for the other methods, as well as the GraphRNN and GRAN baselines. A detailed table illustrating this ablation study will be included in the supplementary material and this property will be clearly stated in the main text.
>
>
> **Out of distribution compression and required sample sizes.** The ability of our graph compressor to transfer to other distributions, was evaluated with two different training datasets, one molecular (MUTAG) and one social (IMDBBINARY). Subsequently, we used the trained model to compress the rest of the datasets and obtained the following results:
>
> |*dataset* | train on MUTAG (bpe) | train on IMDBBINARY (bpe) | train on *dataset* (bpe)|
> |---|---|---|---|
> |MUTAG| - | 6.00| 2.38 |
> |PTC| 4.33 | 6.95 | 2.73|
> |ZINC| **2.90** | 6.15 | *2.19*|
> |PROTEINS| 4.76 | 4.79| 3.14|
> |IMDBBINARY|  1.81 | - | 1.05|
> |IMDBMULTI| 1.36| **0.77** | **0.73**|
>
> *Remark 1: out of distribution generalisation.* As expected, the compressor can generalise to other similar distributions (e.g., MUTAG and ZINC contain molecules of similar sizes, while both IMDBBINARY and IMDBMULTI are movie collaboration egonets).
>
> *Remark 2: learning from few samples.* Another interesting insight we obtained from this experiment is that our method is efficient in terms of the number of samples it requires to be trained. In particular, the MUTAG training set contains only 169 graphs and, strikingly, it generalises to ZINC that contains 10K graphs. In addition to that, we would like to comment here that the datasets we experimented on have training set sizes in the order of magnitudes of 100/1K and 10K graphs, and in every case we easily generalise to unseen data (as can be seen from the results above), hence we can conclude that sample complexity is not an issue.
>
>
> ### 5. Additional comments
>
> * L143: Indeed, the size of the compression model becomes less important when dealing with large datasets. However, in some cases, such as when the model is an overparametrised neural network, the model size is a very large constant that should not be ignored in order to fairly compare different graph compressors. This intuition becomes evident in our experiments, where taking into account the model size renders the total description length of the output larger than the uncompressed data themselves. Please also see our response to R1. Regarding generalisation to unseen data, please see our response above.
>
> * Diagram: Thank you for the suggestion, we agree that having a diagram will make it easier for the reader to obtain high-level understanding. An illustrative figure depicting the overall pipeline will be included in an updated version of the paper to make the method more accessible.
>
> * L90, L94, L124, L247: We are thankful for these suggestions. Appropriate rephrasing and extra explanations will be given in an updated version.
>
> * OGB code: Thank you for your suggestion. Indeed, this is a graph distribution that we have not explored and we would be also curious to observe the dictionary atoms that might arise in this case. Furthermore, an interesting challenge that we would like to address in the future is scaling our partitioning algorithm to datasets with larger graph sizes, where, according to our theoretical results, the compression quality will be even more pronounced.

---

### Official Review · Reviewer_5v1K · 2021-07-18

**Rating:** 8
**Confidence:** 3

**Summary:**

The authors establish a theoretically well-motivated, general-purpose graph compression framework they call Partition-and-Code (PnC).
The framework consists of 3 general parts: partitioning, creating a dictionary of common subgraphs and entropy coding. Crucially, the first two steps are used to map a graph to its isomorphism class, and it is in fact the isomorphism of the input graph that is encoded using entropy coding.

The compression pipeline, given a graph partitioning function and a subgraph dictionary, works as follows:
1) The graph to be encoded is split up into two types of subgraphs: atoms (subgraphs of a fixed size, such that they should be likely to appear in the dictionary), and cuts, which connect the atoms.
2) Then, atoms appearing in the dictionary are encoded using a distribution tailored to the dictionary. Cuts and atoms missing from the dictionary are coded using an appropriate uniform distribution.

Importantly, ML techniques are only used as heuristics for the partitioning part of the encoding process, which means that the decoder does not need to know the model parameters.
The authors propose various proof-of-concept ways to learn the partitioning function using an e.g. GNN and propose a relaxed, greedy gradient-based way to learn the atom dictionary. Learning of all parts of the encoder is performed by minimizing an appropriately defined minimum description length objective.

The authors perform experiments on multiple popular graph datasets and achieve good results.


**Limitations And Societal Impact:**

I believe they addressed the limitations of their work very thoroughly.

**Main Review:**

# Strengths
The work is based on a well-motivated idea and contains several well-executed solutions to the problems considered.

The general framework of PnC is elegant, and by operating on the domain of graph isomorphism classes instead of graphs they already achieve good gains in compression efficiency.

The proposed solutions to all 3 steps are simple, sensible and work well.

The fact that the decoder does not require any statistical model beyond the coding distribution to decode the compressed graph representation is very appealing both in terms of computational as well as run-time costs and makes the method widely applicable.

The experimental methodology is sound. The authors perform comparisons to a large number of other methods on multiple graph datasets and their method achieves very good results on all of them.

The paper is very clearly written with a logical, easy-to-follow layout.

# Weaknesses
My only issue with this otherwise very nice submission is in their comparison with what the authors call "neural, likelihood-based approaches".
The authors assume that some representation of the model that was used to compress the graph has to be transmitted along with the compressed graph representation.
In my opinion, this is a fair assumption, but my issue is that it is assumed that the model parameters are assumed to be uncompressed during transmission (the authors assume 16 bits/parameter).
While of course, it is easier to calculate the model size this way, I think this doesn't provide a representative comparison,
and hence in the current state of the paper, I cannot conclude that the authors' method achieves the best compression among all considered methods. There has been a lot of work recently on the compression of neural networks, with some
recent works (e.g. [1]) achieving up to a 1000-fold reduction in model size. Therefore, I would recommend the authors to perform further experiments where the transmitted neural model is compressed using any reasonable, recent model compression method, include the results in Table 1 and see if their method still performs better.

If the authors can address my concerns, I will be very happy to increase my score.

Additionally, some typos that I found:
- Line 115: Reference 53 appears twice
- Line 338: "... store each each in a dataset ..." - the second "each" should read "edge".

# After the Rebuttal

After reading the other reviewers' comments and the authors responses, I am even more impressed with the work. The authors have addressed my main concern with their work, and therefore I raised my score on the submission.

[1] M Havasi, R Peharz, and JM Hernández-Lobato. Minimal random code learning: Getting bits back from compressed model parameters. ICLR 2019.



**Time Spent Reviewing:**

4-6

---

> ### Author Response · Authors · 2021-08-10
> **Official author reply to Reviewer 5v1K**
>
> To begin with, we would like to thank the reviewer for their thorough evaluation of our work and for pointing out our main contributions. Below we address your concern:
>
> ### Comparison with neural likelihood-based approaches (compressing the model)
>
> *Following the reviewer's recommendation, in the updated version of our manuscript, we will include an additional baseline in Tables 1 and 2, where we will compare against deep generative models that have undergone model compression.* Given that currently, model compression is an active research area and, to the extent of our knowledge, there is no work on how to compress graph generators, we would like to stress that this is a problem at its own sake. Here, we choose one of the most popular methods, i.e., the Lottery Ticket algorithm and iteratively pruned the model weights and report the best total description length (likelihood + model size).
>
> Below, we discuss our methodology and findings in detail:
>
> **Step 1.** *First, we determine the extent to which the neural-likelihood model would need to be compressed to match PnC* (assuming that the compression does not lead to any degradation in the network's ability to model the distribution). The results are as follows:
>
> |dataset|GraphRNN|GRAN|
> |---|---|---|
> |MUTAG| x1264 | x3412
> |PTC| x484|x3173
> |ZINC| x38|x90
> |PROTEINS| x60| x168
> |IMDBB| infeasible | x763
> |IMDBM| infeasible | x1033
>
>
> We make two observations:
>
> (a) In 6 out of the 12 cases it is impossible to outperform PnC with the current state-of-the-art in model compression technology (Ref [1] provided by the reviewer gives up to x1000 model compression - this is the highest compression rate that we have seen in the literature). Interestingly, in some cases (graphRNN for IMDBB and IBDMM) the likelihood computed by the deep generative model is smaller than PnC, hence it is impossible to outperform PnC independently of how good the compressor is.
>
> (b) As perhaps expected, the model size becomes less important as the dataset size grows (i.e., ZINC and PROTEINS). The latter is particularly pertinent for datasets with a high degree of self-similarity, such as small molecules, as the DNN can obtain a good estimation of the underlying distribution. These are the cases for which we will perform our comparative study (the experiments were performed on graphRNN since its model size was smaller).
>
> **Step 2.** *Second, we decrease the model size by hyper-parameter search.* Since even the smallest required compression rates (x38, x60) are moderately large even for pruning techniques, we performed a hyperparameter search in terms of model width and depth, in order to further reduce the size of the original network without heavily compromising the likelihood. This led to ~ x10 reduction in the model size (~30K parameters).
>
> **Step 3.** *Finally, a lottery ticket iterative pruning was performed.* The model size was again calculated using 16bits/parameter and the following formula for storing the non-zero entries of each parameter matrix in $\mathbb{R}^{d_1 \times d_2}$ was used: $\log (d_1 d_2 + 1) + \log {d_1 d_2 \choose  e}$, where $e$ the number of non-zero entries. We assumed that the architecture is public and, thus, does not need to be transmitted.
>
> |dataset|data (bpe) | total (bpe)
> |---|---|---|
> |ZINC| 1.72 | 1.94
> |PROTEINS| 2.89 | 4.60
>
> As observed, the total description length is indeed significantly reduced. Moreover, in the ZINC dataset, the compressed graphRNN slightly outperforms PnC. We make the following remarks:
>
> 1. Whenever deep generative models can faithfully estimate the underlying density, model compression seems to be a promising research direction in order to obtain parsimonious compressors, competitive to PnC. As we see in our preliminary experiments, network pruning might be able to preserve the accuracy of a graph generative model. The examination of this topic is ripe for future investigation.
>
> 2. *No explicit control of the description length*.  Model compression can be rather heuristic/tedious and, unlike the PnC framework, it does not account for the total description length (this is why in our study we had to manually search the hyperparameter space for smaller models). Some recent efforts, such as Ref [1] that the reviewer suggested, attempt to deal with this issue. This is another interesting direction for neural compression.
>
> 3. *Model compression does not always suffice to beat PnC*. As observed, the ability of PnC to obtain good likelihood estimates together with the guarantee for a small number of parameters makes it a tough-to-beat competitor for graph compression.
>
> We thank the reviewer for raising this excellent point. The above discussion will be added to an updated version of our manuscript. Moreover, the preliminary results reported here will be extended to all the datasets, as well as for the GRAN baseline and will be added to the main table.
>
> [1] M Havasi, R Peharz, and JM Hernández-Lobato. Minimal random code learning: Getting bits back from compressed model parameters. ICLR 2019.

---

### Author Response · Authors · 2021-08-10
**Official reply to all reviewers: Theoretical analysis - the compression gains of PnC**

### 1. Theoretical analysis


Reviewer u2r2 and Reviewer Q2XE expressed interest in a deeper theoretical study of the benefits of PnC. In the supplementary material, we provided an initial argument regarding the importance of subgraph isomorphism (Appendix A). This analysis can be further extended by theoretically comparing important graph compressors families (complementing our experimental results of Tables 1 and 2) with PnC. In particular, under mild assumptions (that fit nicely with real-world graph distributions) we can derive an asymptotic total ordering of their compression quality. Below follows a brief summary:

We compare PnC against two strong baselines:

(a) *Pure partitioning-based graph encodings* (code length denoted as $\mathrm{L}_{\text{part}}(G)$). Here, a graph is decomposed into subgraphs and cuts, but the distribution of the subgraphs is not modelled, i.e., we do not use a dictionary and everything is encoded with the help of a null model.

(b) *Null model-based encodings* that do not rely on partitioning but encode each graph as a whole ($ \mathrm{L}_{\text{null}}(G)$). This family includes the null models in Tables 1 and 2, most notably the labelled Erdos-Renyi model (Eq. (2) in the main paper), but also the unlabelled counterpart that asymptotically saves $\Theta (n \log n)$ bits (this is what the SZIP algorithm approximates, ref [9]).

**Theorem 1 (informal).** Consider a distribution $p$ over graphs with $n$ vertices and a partitioning algorithm that decomposes a graph into $b$ blocks of $k = O(1)$ vertices. Then, under mild conditions (Eq.(1), (2) below),  it holds that:

\begin{equation}
\mathbb{E}_{G \sim p}[L_\{PnC\}(G)]
     \stackrel{(1b)}{\lesssim}
\mathbb{E}_\{G \sim p\}[L_\{part\}(G)]
     \stackrel{(1a)}{\lesssim} \mathbb{E}_\{G \sim p\}[L_\{null\}(G)]
\end{equation}
The absolute compression gains are:
\begin{equation}
    \mathbb{E}_\{G \sim p\}[L_\{part\}(G)]  \lesssim \mathbb{E}_\{G \sim p\}[L_\{null\}(G)] - \Theta(n^2) \text{ and }
 \mathbb{E}_\{G \sim p\}[L_\{PnC\}(G)] \lesssim \mathbb{E}_\{G \sim p\}[L_\{part\}(G)]
    - \Theta(n)
\end{equation}


**Proof sketch.** We focus on directed graphs. For all the algorithms we give an asymptotic growth rate of the compression quality assuming that the number of edges in a graph $m$ grows at the same rate as the maximum number of edges $n^2$, i.e., $m = c n^2$ where $c$ is small constant $<1$. We employ the following approximation: $\log{n \choose m} \approx n \textrm{H}(\frac{m}{n})$, where ${\textrm{H}(p) = -p\log p - (1-p)\log(1-p)}$ is the binary entropy of a Bernoulli variable with probability of success $p$.

We remind that $0\leq \textrm{H}(p)\leq 1$, where the l.h.s equality is satisfied for $p \in \{0,1\}$ and the r.h.s. for $p=\frac{1}{2}$.

- *Null models*: For the best null model (unlabelled Erdos-Renyi), the description length follows from Eq. (2) after subtracting $\Theta( n\log n)$ bits, hence the growth rate is quadratic and in particular it is $\log {n^2 \choose m} \approx n^2 \textrm{H}(\frac{m}{n^2})$, where we used the binary entropy approximation.

- *Pure partitioning*: This is equivalent to encoding the adjacency matrix with (a) $b = \frac{n}{k}$ diagonal blocks, i.e., the subgraphs of size $k^2$, and with (b) $b^2-b = \frac{n^2}{k^2} - \frac{n}{k}$ off-diagonal blocks also of size $k^2$, i.e., the cuts. When $k = O(1)$, it is easy to see that the dominating term in the description length are the off-diagonal elements (since it grows with $n^2$). Using a simplified encoding model for the cuts similar to Eq. (1) in the appendix, the description length can be shown to grow with $n^2\big(\frac{\log(k^2 + 1)}{k^2} + \bar{\textrm{H}}(\frac{m_{ij}}{k^2})\big)$, where $\bar{\textrm{H}}(\cdot)$ is the expected value of $\textrm{H}(\cdot)$ and $m_{ij}$ the size of the cut between two subgraphs. The diagonal block elements (i.e., the subgraphs) contribute to the description length a linear term equal to  $\frac{n}{k}\big(\log(k^2 + 1) + k^2\bar{\textrm{H}}(\frac{m_i}{k^2})\big)$ bits (encoding the edges $m_i$ of each subgraph, as well as their positions).

- *PnC*: In the case of PnC with a dictionary, the cuts are encoded in the same way as in the pure partitioning compressor. The description length of the diagonal block elements, as compared to pure partitioning, largely depends on the probability 1 - $\delta$ with which a subgraph is encoded as a dictionary atom. PnC will need (a) $\frac{n}{k}\delta\big(\log(k^2 + 1) + k^2\bar{\textrm{H}}(\frac{m_i}{k^2})\big)$ bits for non-dictionary subgraphs and (b) $\frac{n}{k}(1 - \delta) \mathbb{H}(D)$ bits for dictionary subgraphs, where $\mathbb{H}(D) = \mathbb{E}_{a}[-\log q_\phi(a)]$ is the expected description length of a dictionary atom.

Now we can derive the conditions for the asymptotic inequalities of Theorem 1 by contrasting the growth rates of the description lengths as established above. In particular, contrasting the quadratic terms, follows the condition of theorem (1a):
\begin{equation}
    \frac{\log(k^2 + 1)}{k^2}) +\bar{\textrm{H}}\bigg(\frac{m_{ij}}{k^2}\bigg) < \bar{\textrm{H}}\bigg(\frac{m}{n^2}\bigg),
\end{equation}
while contrasting the linear terms and using the fact that $\mathbb{H}(D) \leq \log|D|$, follows the condition of theorem (1b):
\begin{equation}
    |D| <  (k^2 + 1)2^{k^2\bar{\textrm{H}}\big(\frac{m_i}{k^2}\big)},
\end{equation}

Eq. (1) says that as long as there exists a “low-surprise” clustering structure in the graph (e.g., subgraphs are either densely or sparsely inter-connected), then $\bar{\textrm{H}}(\frac{m_{ij}}{k^2})$ will be small and partitioning-based compressors will always incur quadratically less bits. Since clusterability (e.g., community structure in social networks) is common in real-world networks this is a rather mild assumption.



Eq. (2) implies that even if the dictionary is relatively large, PnC will always incur linearly less bits than pure partitioning. This confirms our intuition that we need to encode subgraphs as isomorphism classes instead of adjacency matrices (this is corroborated by the theorem in Appendix A).
We would like to direct the attention of the Reviewer u2r2  here, since this provides theoretical evidence related to their concern w.r.t. importance of the dictionary.



### 2. Partitioning algorithm in the main text

As advised by Reviewer dgax, Reviewer u2r2 and Reviewer Q2XE, a more detailed presentation of the partitioning algorithm (perhaps including Algorithm 1, P5 in the appendix) will be given in the main text and the technical details will be kept in the supplementary material. Given the space constraints, we chose to differ it to the supplementary in order to emphasise the general formulation of the framework, since various different algorithmic solutions can be proposed for (learnable or not) partitioning. However, we agree that this part of the approach is important and non-trivial and therefore more focus will be given.

---

### Decision · Program_Chairs · 2021-09-27

**Decision:**

Accept (Poster)

**Comment:**

The paper presents learning-based method for compressing graphs. The method proceeds by partitioning the input graph into simpler parts, and using a learned dictionary to encode frequently appearing subgraphs. Empirical evaluation demonstrates improvements over multiple baselines.

The main concern was that the sizes of graphs used in evaluation were relatively small. However, given that the technique was shown to generalize to several diverse data sets, most reviewers were comfortable that the results should generalize to larger graphs as well.